# Early line and hook fishing at the Epipaleolithic site of Jordan River Dureijat (Northern Israel)

**Antonella Pedergnana[1]◉\*, Emanuela Cristiani[2]◉\*, Natalie Munro[3]◉, Francesco Valletta[4]◉, Gonen Sharon**  [5]◉

**1** TraCEr, Laboratory for Traceology and Controlled Experiments at MONREPOS Archaeological Research Centre and Museum for Human Behavioural Evolution, RGZM, Mainz, Germany, **2** DANTE–Diet and Ancient Technology Laboratory, Department of Oral and Maxillo-Facial Sciences, Sapienza University of Rome, Roma, Italy, **3** Department of Anthropology, University of Connecticut, Storrs Mansfield, Connecticut, United States of America, **4** Institute of Archaeology, The Hebrew University of Jerusalem, Jerusalem, Israel, **5** Department of Galilee Studies (M.A.), Tel Hai College, Qiryat Shemona, Israel

◉ These authors contributed equally to this work.
\* antonella.pedergnana@gmail.com (AP); emanuela.cristiani@uniroma1.it (EC)

## Abstract

Nineteen broken and complete bone fish hooks and six grooved stones recovered from the Epipaleolithic site of Jordan River Dureijat in the Hula Valley of Israel represent the largest collection of fishing technology from the Epipaleolithic and Paleolithic periods. Although Jordan River Dureijat was occupied throughout the Epipaleolithic (~20–10 kya the fish hooks appear only at the later stage of this period (15,000–12,000 cal BP). This paper presents a multidimensional study of the hooks, grooved stones, site context, and the fish assemblage from macro and micro perspectives following technological, use wear, residue and zooarchaeological approaches. The study of the fish hooks reveals significant variability in hook size, shape and feature type and provides the first evidence that several landmark innovations in fishing technology were already in use at this early date. These include inner and outer barbs, a variety of line attachment techniques including knobs, grooves and adhesives and some of the earliest evidence for artificial lures. Wear on the grooved stones is consistent with their use as sinkers while plant fibers recovered from the grooves of one hook shank and one stone suggest the use of fishing line. This together with associations between the grooved stones and hooks in the same archaeological layers, suggests the emergence of a sophisticated line and hook technology. The complexity of this technology is highlighted by the multiple steps required to manufacture each component and combine them into an integrated system. The appearance of such technology in the Levantine Epipaleolithic record reflects a deep knowledge of fish behavior and ecology. This coincides with significant larger-scale patterns in subsistence evolution, namely broad spectrum foraging, which is an important first signal of the beginning of the transition to agriculture in this region.

**Data Availability Statement:** All archaeological materials (fish hooks, grooved stones, fish bones) are curated in the Department of Galilee Studies at Tel Hai College, Israel. All relevant data on the fish

hooks and grooved stones are published within the paper. An inventory of the bone hooks and grooved stones including catalog numbers are indicated in Table 1 and Table 3. The fauna are too numerous to list individually by number, but the data presented in Figure 14 are available in an online repository [https://doi.org/10.7910/DVN/VTKRQW] as are the 3D models of the hooks and grooved stones.

**Funding:** Research is supported by the Israel Science Foundation - www.isf.org.il (Grant #918/17) granted to G. Sharon. This research was also supported by grants from the European Research Council - https://erc.europa.eu (Starting Grant Project HIDDEN FOODS, grant no. 639286) awarded to E. Cristiani and the National Science Foundation - https://www.nsf.gov (BCS-1842087) awarded to N. Munro. The funders had no role in study design, data collection and analysis, decision to publish, or preparation of the manuscript.

**Competing interests:** The authors have declared that no competing interests exist.

# Introduction

Fish remains first appear in hominin sites close to two million years ago [1]. Although fish skeletal remains and isotope data provide reasonable proxies for the role of fish in hominin diets [2], the technology used for their capture in prehistory is more elusive. This is primarily because the vast majority of fishing gear was made from perishable materials such as wood and plant fibers that rarely survive in archaeological contexts.

Over the last ten years, a number of exceptionally well-preserved bone fish hooks and grooved stones, were recovered from the waterlogged Epipaleolithic site of Jordan River Dureijat (JRD) on the Upper Jordan River in the Hula Valley of modern-day Israel. The fishing gear from JRD is unique in a number of dimensions that contribute significantly to our understanding of technological and subsistence evolution at the end of the Pleistocene. First, although earlier shell hooks have been found in Southeast Asia [3], the hooks originating from the Natufian layers of JRD (15,000–12,000 cal BP; [4] are the largest single dataset on line fishing found in Southwest Asia and Europe up to this point in time. Thus far, the assemblage includes 19 complete hooks and hook fragments. The hooks come in a variety of forms and sizes and several display informative features such as barbs, grooves, residues and wear. In addition, the hooks are found in the same archaeological context as an assemblage of grooved pebbles that may have served as sinkers in a very early, but sophisticated line and hook technology. The JRD assemblage thus provides an unusual opportunity for a multidimensional study of the manufacture and use of early line and hook technology and its relationship to human subsistence evolution.

We achieve these goals using, a multidisciplinary approach that presents and synthesizes archaeological and experimental observations with the results of four specialized analyses. These include a technological study of the hooks based on accurate measurements of hook linear dimensions and angles and the presence and absence of key features using 3D scanned images, high magnification studies of the hooks and grooved stones to detect evidence of manufacture or use-wear and residues and a preliminary zooarchaeological study of the fish remains at the site. Ultimately, these results are combined to reconstruct hook and line fishing at JRD and to situate it within the broader regional context of subsistence change at this important moment in prehistory. In particular, the appearance of this technology in the Levant coincides with the emergence of broad spectrum resource use in the periods leading up to the beginning of agriculture.

## Early fishing

The earliest record of fish at a hominin site comes from the Koobi Fora Formation in Kenya and dates to 1.95 million years ago [1]. Since these early remains were discovered, archaeologists have paid closer attention to the use of fish in anthropogenic sites, identifying fish remains or indirect evidence for fishing based on use wear traces or residues on stone tools from a number of Lower and Middle Paleolithic and Middle Stone Age sites in Africa and Eurasia [5–10]. Marean [9] argues that the earliest evidence for a coastal adaptive strategy appears with early *Homo sapiens* about 110 kya in South Africa. Although the presence of fish is increasingly apparent in these early sites and others, they are not common and even if they were eaten, fish do not become substantial or routine components of human diets until much later.

Fish take on increasingly important roles in the Upper Paleolithic economies in east and south Asia after this date [3,11]; and increase in ubiquity and abundance throughout the Upper Paleolithic in other parts of Europe and Asia as well [12–15]. Direct evidence for the exploitation of coastal Mediterranean species and cold-adapted Atlantic species appears in the

Solutrean at Cueva de Nerja in Spain [16,17]. Relief sculpture and painting as well as body adornments, from Upper Paleolithic sites in Europe also reveal that fish became sufficiently important to hold a role in forager imaginations. The significance of fish remains increases in Epipaleolithic and Mesolithic sites, especially those situated in coastal, lake or riverine settings in Northern Europe and the Atlantic [18]. Specialized fishing activities are also well documented in the Mediterranean during this period at Vela Spila in Croatia [19], Franchthi Cave in Greece [13,20] and Grotta dell'Uzzo in Italy [21]. In southeastern and northern Europe, an increase in the role of freshwater fish in the diet of forager populations is attested from the early Mesolithic onward on the basis of fish remains and stable isotope values in bone collagen [22–29], as well as in dental calculus [27].

## Fishing technology

Despite the interest in the time depth of aquatic resource exploitation and broad spectrum human diets, fishing technology is elusive throughout the Paleolithic. While fishing technology may have included a wide range of tools such as traps, lines, gorges, weights, barbed points, harpoons and hooks, such artefacts are rarely recovered from ancient prehistoric sites. Barbed points and harpoons appear early on in Africa [30] and were important cultural markers of Upper Paleolithic techno-complexes such as the Magdalenian and/or the Azilian [31,32] but their association with fishing is not secure [33,34]. Other fishing technologies such as nets or traps, are made largely of perishable materials that preserve only in unusual conditions, thus their early appearance is difficult to pinpoint in the archaeological record. Indisputable fishing technology like hooks do not appear until the end of the Upper Paleolithic and the Epipaleolithic when zooarchaeological evidence for fishing also expands in earnest as part of a larger shift toward the exploitation of smaller game species as diets diversify.

Possible straight fish hooks (gorges or *hameçon droits*) were in use during the Magdalenian when artistic representations of fish appear [35], but the earliest fish hooks known thus far are made of shell and come from the cave of Asitau Kuru (formerly known as Jerimalai) in East Timor [36]. A broken shell-fish hook from the site has been dated between 16 and 23 kya [3]. Another early fish hook, also made from shell, was found at Sakitari Cave in Japan (22,380 −22,770 cal BP) [11]. Archaeological excavations at the site of Makpan Cave have also recovered entire hooks, roughouts and preforms, exposing the entire chaine operatoire for the production of shell hook. The earliest bone fish hooks, date to Late Epipaleolithic (Natufian) sites in the southern Levant [37] and include those from JRD presented here. After this, bone fish hooks become widespread. Slightly more recent hooks have also been found in the Late Upper Paleolithic deposits at the site of Wustermark 22 in northeast Germany (ca. 12,300 cal BP) and are frequently found in Mesolithic sites in Europe [38,39]. Later in the Mesolithic, waterlogged deposits where anaerobic conditions allowed the preservation of otherwise perishable botanical remains, are a primary source for prehistoric fishing gear, including fish-traps, nets, gorges, hooks, harpoons, weirs and more [25,28,29,40–42].

## Early fishing and fishing technology in the Epipaleolithic southern Levant

In the southern Levant abundant fish remains were found at the site of Gesher Benot Ya'aqov located on the banks of the Upper Jordan River within 1.5 kilometer of JRD [7,43,44]. Fish bones make sporadic appearances in Middle and Upper Paleolithic sites in the Levant, but they are always rare [45]. Indisputable evidence for invested human fishing first appears at Early Epipaleolithic Ohalo II (ca. 23 cal kya) on the shore of the Sea of Galilee [46–49]. The fish bone assemblage is accompanied by material remains that probably served as fishing

technology such as net sinkers [50,51]. Importantly, fragments of twisted fibers were preserved in the waterlogged sediments and may have been used to make nets or traps [52].

After this time, fish are much more ubiquitous in Levantine archaeological sites [53], especially in sites close to major water bodies such as the Mediterranean Sea and the freshwater system of the Jordan Valley [54]. Fish are found in most Natufian sites with good preservation and careful recovery [4,37,45,50,53,55], but are especially abundant at Natufian sites in the Upper Jordan Valley including Eynan in the Hula Valley [56,57] and Nahal Ein Gev II near the eastern shore of the Sea of Galilee [58].

The Early Natufian layer at Kebara Cave was home to the first fish hooks found in the Levant, as well as harpoons that may have been used for fishing. Two complete hooks, one plain and one grooved around the neck, and two broken hooks were recovered [37,59]. Numerous bone fish hook fragments were also found in the Final Natufian layer at Eynan [57,60] as was one broken hook at Early Natufian Hof Shahaf near the northwestern corner of the Sea of Galilee [61]. A single hook was found at Late Natufian Hayonim Terrace [62]. After the Natufian, the use of fish hooks continues into the Pre-Pottery Neolithic, although primarily at Mediterranean coastal sites. They are especially common at the water-logged site of Atlit Yam [63–65]. The switch to metal as the primary raw material for fishing hooks production took place as early as the Chalcolithic period in the Levant [66].

## The site of Jordan River Dureijat (JRD)

Jordan River Dureijat (JRD) is an ephemeral, short-term encampment located on the southern shore of Paleolake Hula that was intermittently occupied over a span of about 10,000 years (ca. 20,000 to 10,000 cal BP) during the Levantine Epipaleolithic period [4]. Stratigraphic, radiocarbon and technological evidence indicate that JRD is one of very few sites to be occupied in each of the Early, Middle and Late Epipaleolithic phases.

JRD is located on the east bank of the Upper Jordan River where it flows south out of the Hula Valley in northern Israel (Fig 1). The site was discovered during an archaeological survey preceding a massive drainage operation of the Jordan River in 1999 [67,68]. JRD's archaeological horizons stretch across 50 m of the river bank adjacent to the outlet of the small Dureijat Stream (Fig 1). A test excavation undertaken in 2002 [69] was followed by six full excavation seasons between 2014 and 2019. Excavation exposed a sequence of sediments that accumulated within the fluctuating water levels of Paleolake Hula. The archaeological horizons are found within near-shore layers that accumulated during low water stands and reflect human activity on the bank of Paleolake Hula. These archaeological horizons are separated by archaeologically sterile, fine silt layers deposited when the lake level was higher (Fig 2).

The site's stratigraphic sequence and chronology were established in Area B, the primary excavation area at the site (Fig 2D). The chronology was built from numerous radiocarbon dates (Fig 2; [4]. The lowest layer excavated at the site (Layer 5) dates to ca. 20,000 years cal BP and is affiliated with an Early Epipaleolithic lithic tradition. The lithics from Layer 4 are clearly ascribed to the Middle Epipaleolithic, Geometric Kebaran tradition and date to 17,460–15,750 cal BP. The three upper, layers 3c, 3b and 3a, are affiliated with the Late Epipaleolithic Natufian culture (15,000–11,500 cal BP) based on radiocarbon dates and the presence of typical flint artifacts such as microlithic lunates [4,70]. The archaeological sequence of the site ends with silty Layer 3–0, dated to the initial Holocene. Because the accumulation of Layer 3–0 is complex, and the anthropogenic input is not as significant as in other archaeological layers, the context is less secure than the archaeological horizons below. Nevertheless, the finds from this layer include artifacts such as El-Khiam arrowheads and limestone axes that can be clearly ascribed to the Early Neolithic.

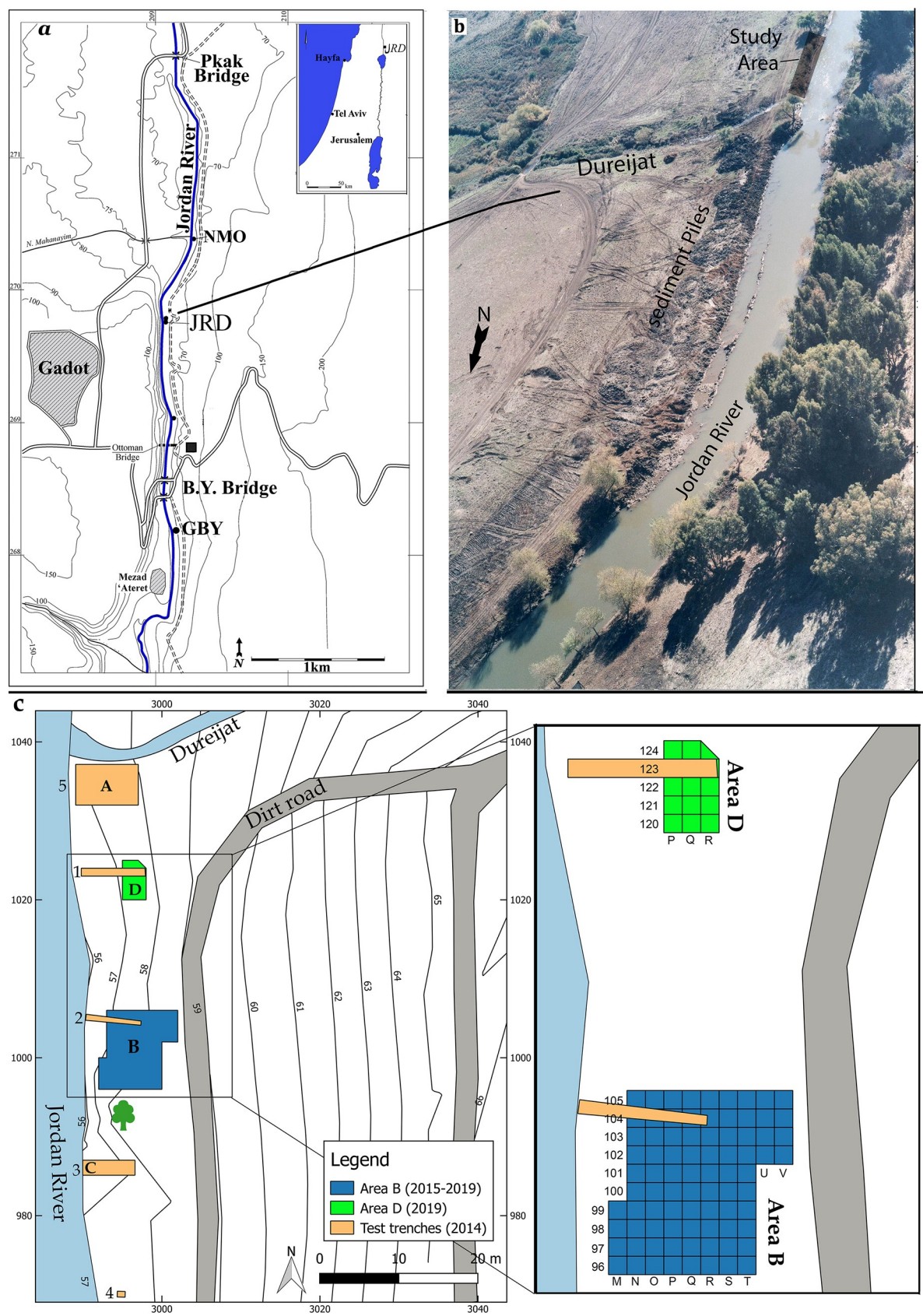

**Fig 1.** a) Location of the site of JRD; b) view of JRD from the east and its location on the Jordan River adjacent to the Dureijat stream; c) map of excavation areas and test trenches.

During the 2019 season, an additional test excavation (Area D) was opened 10 meters to the north of Area B (Fig 1D). This area yielded lower artifact densities, likely because of its original location in the nearshore, shallow water of the paleolake itself. Nevertheless, in addition to faunal and lithic remains, three bone fish hooks and a single grooved pebble were excavated from this area. The stratigraphy of Area D has been correlated with the stratigraphy in Area B (Fig 2).

The archaeological evidence from JRD indicates that although some of the intermittent visits by Epipaleolithic peoples were longer and involved more intensive activities especially during the Natufian period, the excavated area was never used for habitation. JRD was a place that people visited again and again to take advantage of the confluence of diverse lake shore resources. Like other archaeological sites on the banks of the Upper Jordan River [71–73], the sediments at JRD have been waterlogged since they accumulated. The rare outstanding preservation of organic remains sets the site apart from most other Levantine Epipaleolithic sites [4,69,74].

## The osseous fish hook assemblage

Nineteen worked bones shaped as fish hooks were unearthed at JRD (Table 1; Figs 3 and 4). Seven are complete (hooks#1, 2, 3, 4, 11, 12 and 13), one of these is broken in two (#11), but both parts were recovered and refitted (hook #13; Fig 3). The other nine hooks are fragments representing different parts of the hook as specified in Table 1.

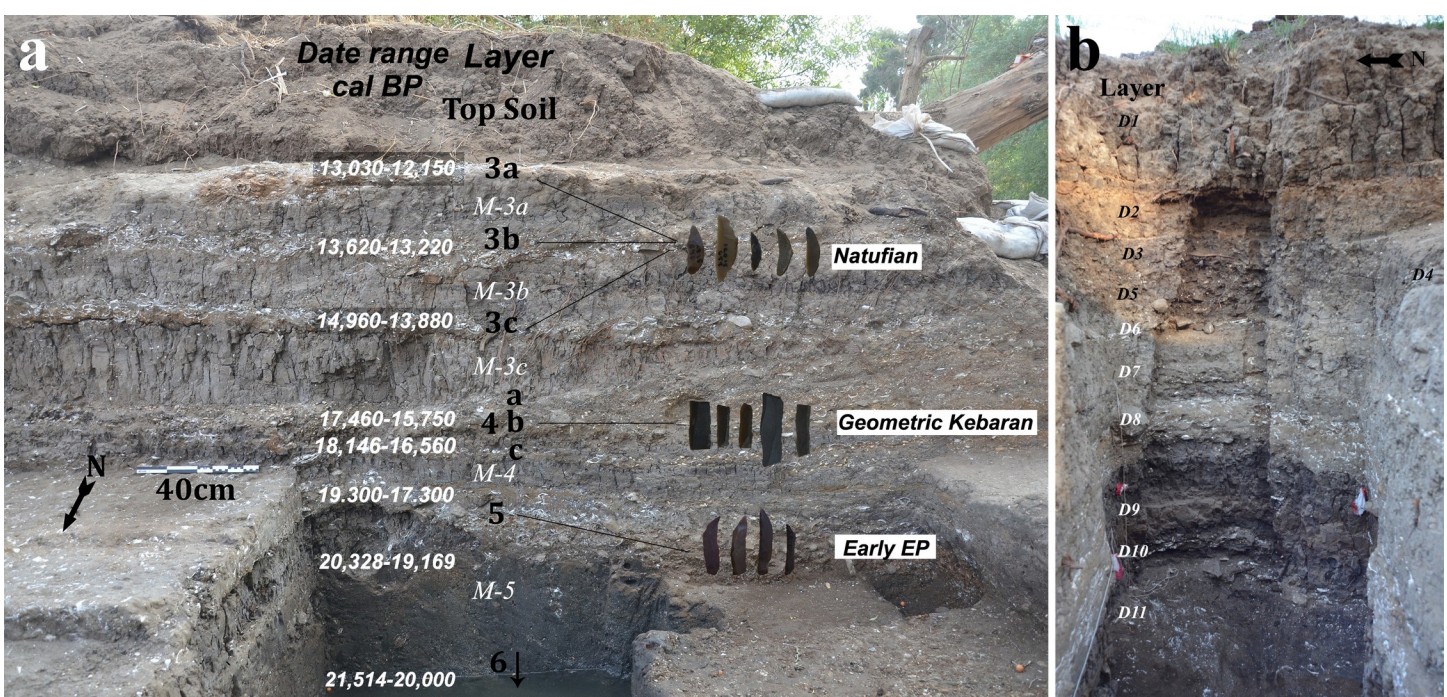

**Fig 2.** a) Stratigraphy, chronology and cultural affiliation of Area B sequence; b) Area D stratigraphic sequence.

**Table 1. JRD fish hook inventory including the presence and absence of hook features.** The JRD hooks are reposited at the Prehistory Laboratory, Tel Hai College, Israel. The 3D fishhook models are available (wrl format) at https://doi.org/10.7910/DVN/VTKRQW.

| Hook # | Layer | Shape | Barb | Line Attachment | Bend Profile |
|---|---|---|---|---|---|
| 1 | 3b | Semi-circular | Inner point barb & outer frontal bend barb | Knob at central shank | Flat and curved |
| 2 | 3b | Typical J | Outer frontal bend barb | Knob at central shank and single groove at top of shank | Round |
| 3 | 3b | Short J | Outer frontal bend barb | Double knob at end of shank | Flat and curved |
| 4 | 3a | Typical J | No | Groove at shank edge | Flat and wide |
| 5 | 3b | Unknown | Outer point barb | Unknown | Flat and thick |
| 6 | 3a | J | Unknown | Knob at very end of shank | Flat and curved |
| 7 | 3a | Short J? | No | Unknown | Round |
| 8 | 3b | Unknown | Unknown | Unknown | Round and curved |
| 9 | 3a | Unknown | No | Unknown | Flat and curved |
| 10 | 3c/4? | Unknown | No | Unknown | Round and thick |
| 11 | 3a | Short J | No | Double groove at shank end | Flat |
| 12 | D3 | Circular | No | Groove at shank end creating a knob above | Round |
| 13 | D8 | J | No | No | Round and wide |
| 14 | 3a | J? | No | Unknown | Flat and thick |
| 15 | 3a | Unknown | Straight angle of point to bend–like frontal lower barb | Unknown | Round |
| 16 | 3a | Unknown | Straight angle of point to bend–like frontal lower barb | Unknown | Flat |
| 17 | D4 | Unknown | Straight angle of point to bend–like frontal lower barb | Unknown | Flat |
| 18 | 3a | Unknown | Unknown | Unknown | Round |
| 19 | 3a | Too fragmented | Unknown | Unknown | Unknown |

Unknown is used for broken hooks where the relevant part of the hook is missing.

### Stratigraphic provenance of the JRD hooks

All of the hooks were recovered from *in situ* excavation contexts with known provenience and stratigraphic location (Table 1; Fig 4). All but two hooks (#7 and 17, Table 1), were found during the sorting of sediments from these contexts after they were wet sieved through 2 mm mesh in the Jordan River. Three broken hooks from Area B (#15, 16 and 18), and one complete hook from Area D (#12), were found at the contact between Layer 3–0 and 3a (Fig 2). Based on the radiocarbon chronology and associated Early Neolithic archaeological finds, Layer 3–0 is likely Early Neolithic in age [4]. The upper surface of 3a may have been disturbed by humans after it accumulated. The similarity of the four hooks to the Natufian assemblage suggest that they derive from the very end of the Epipaleolithic, but they could also belong to the very early Neolithic.

The rest of the JRD bone hooks were found in the three Natufian layers at the site (3a, 3b and 3c), or their stratigraphic equivalents in Area D (Fig 2; Table 1). Six hooks were excavated from Layer 3a and one from the Area D equivalent, Layer D4. Three hooks (#1, 2 and 5) were found in Layer 3b and a single hook, the largest (#13; Figs 3 and 4), was excavated from Layer D8 –which is stratigraphically equivalent to Layer 3c in Area B. A single broken hook (#10) was recovered from the west part of Area B (Square M98) where Layer 3c cuts into Layer 4. In this area [4], the stratigraphy becomes more complex and the separation of the layers is

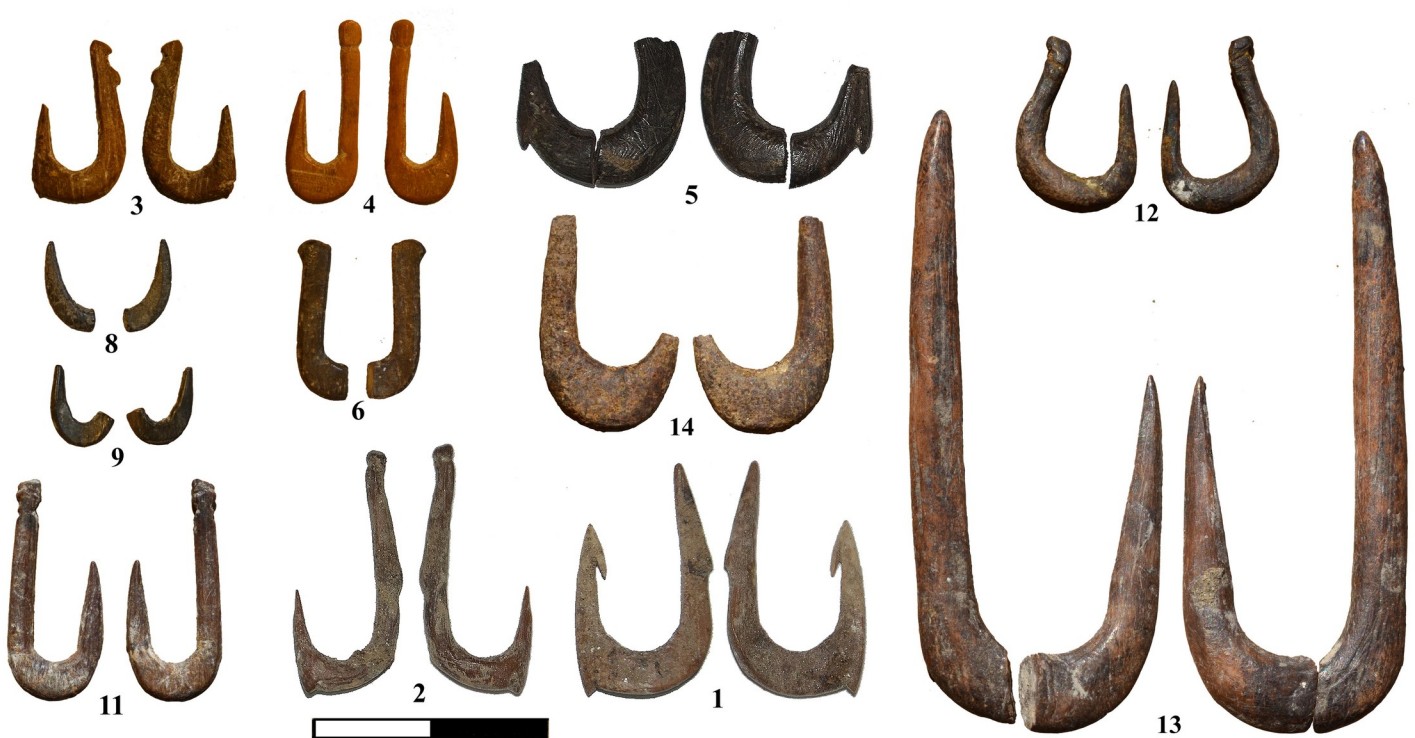

**Fig 3. JRD bone fish hooks.**

challenging and thus the stratigraphic affiliation of hook #10 is the least secure. It is likely that it belongs to Natufian Layer 3c rather than Middle Epipaleolithic Layer 4. It should be noted that Layer 3c and Layer 4 were excavated across a large areal surface (>35 meters; Fig 1D) and all sediments were sieved and sorted. Still, neither layer yielded any hooks other than the large

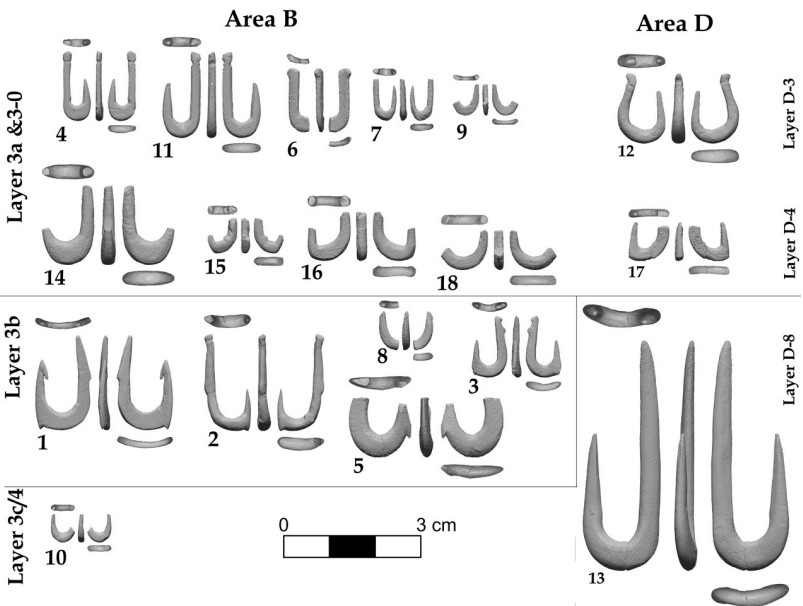

**Fig 4. Images of all JRD bone fish hooks obtained from 3D models, arranged by layer.**

and morphologically divergent hook #13. Hooks were also absent from Layer 5, the oldest layer of the site dating to the Early Epipaleolithic.

In comparison to other archaeological horizons at JRD, Layer 3b is thin, with a low density of archaeological finds. It probably represents a short period of non-extensive human activity in near-shore shallow water. In accordance, the volume of sediments from this layer was small and thus sediments were only sampled for sieving and sorting. Nevertheless, three of the best preserved and unique hooks were unearthed from this layer (Table 1). In contrast, Layer 3c, which encapsulates the earliest stage of the Natufian represented at JRD, is a much more clearly defined archaeological horizon that contained diverse archaeological remains (flint tools, limestone net sinkers, basalt tools, animal bones and botanical remains) compared to other layers, yet it yielded only a single bone tool that can be classified as a hook, but clearly deviates from all other hooks in size and morphology.

## Preservation and raw material

Like the faunal remains from JRD, the state of preservation of the hooks is good overall (Fig 3). Observations under high magnification revealed post-depositional alteration, such as soil and salt concretions, bacteria damage, rootlet etching and sediment patination on a few specimens, but were not severe enough to affect the functional interpretation of the hooks.

All of the hooks from JRD are made of bone except for one fragmentary hook (#16) made of tooth enamel (likely wild boar tusk). Heavy working and reshaping of the bone during hook manufacture erased any diagnostic features on the bones making it challenging to determine the taxon of the bones used to produce the JRD hooks. Analytical methods that can identify taxon based on protein identification are destructive so could not be used in this study. The size and the natural curvature of the hooks, as seen in the section view of hooks #1, 3 and 9 (Fig 4), suggest that some were produced from the long bones of small and medium sized herbivores, such as gazelle or fallow deer, the two most common ungulates in the JRD assemblage.

## Terminology and typology of fish hooks

The terminology used to describe the parts of the fish hook is largely standardized for modern hooks [75,76], although the terms for some parts vary by region and manufacturer (i.e., some prefer the term tip over point or bite over throat). Here, we largely follow the terminology of Thomas et al. [77] developed for contemporary hooks (Fig 5). Nevertheless, because prehistoric hooks sometimes differ from modern hooks in certain features, we also use terminology developed to describe prehistoric and historic fish hooks from the Pacific region [78,79].

The morphology and specific attributes of the fish hooks provide crucial information about their function. Determining the size of certain dimensions and the range of hook attributes is necessary to assess change in the function of the hooks over time, the type of fish targeted by the prehistoric anglers and their fishing techniques. For example, the length, width, and bite size (Fig 5) limits both the size of the bait and the size of the fish that can be captured (small hooks can capture large and small fish, whereas large hooks can capture only large fish). Likewise, the size of the gape, bite and point-angle affect how easily a fish can swallow the hook and how likely the point is to get stuck in the fish's mouth [e.g., 80]. The width of the cross-section of the bend influences the ability of the hook to withstand torque without snapping during fishing. The measurements can thus, provide some insight into the fishing strategies and fish types targeted by the fishers from JRD.

## Measurement of the fish hooks

Studies of prehistoric fish hooks are often regional in character and based on small assemblages [e.g. 39,81–85]. In addition, artifact features are often described qualitatively,

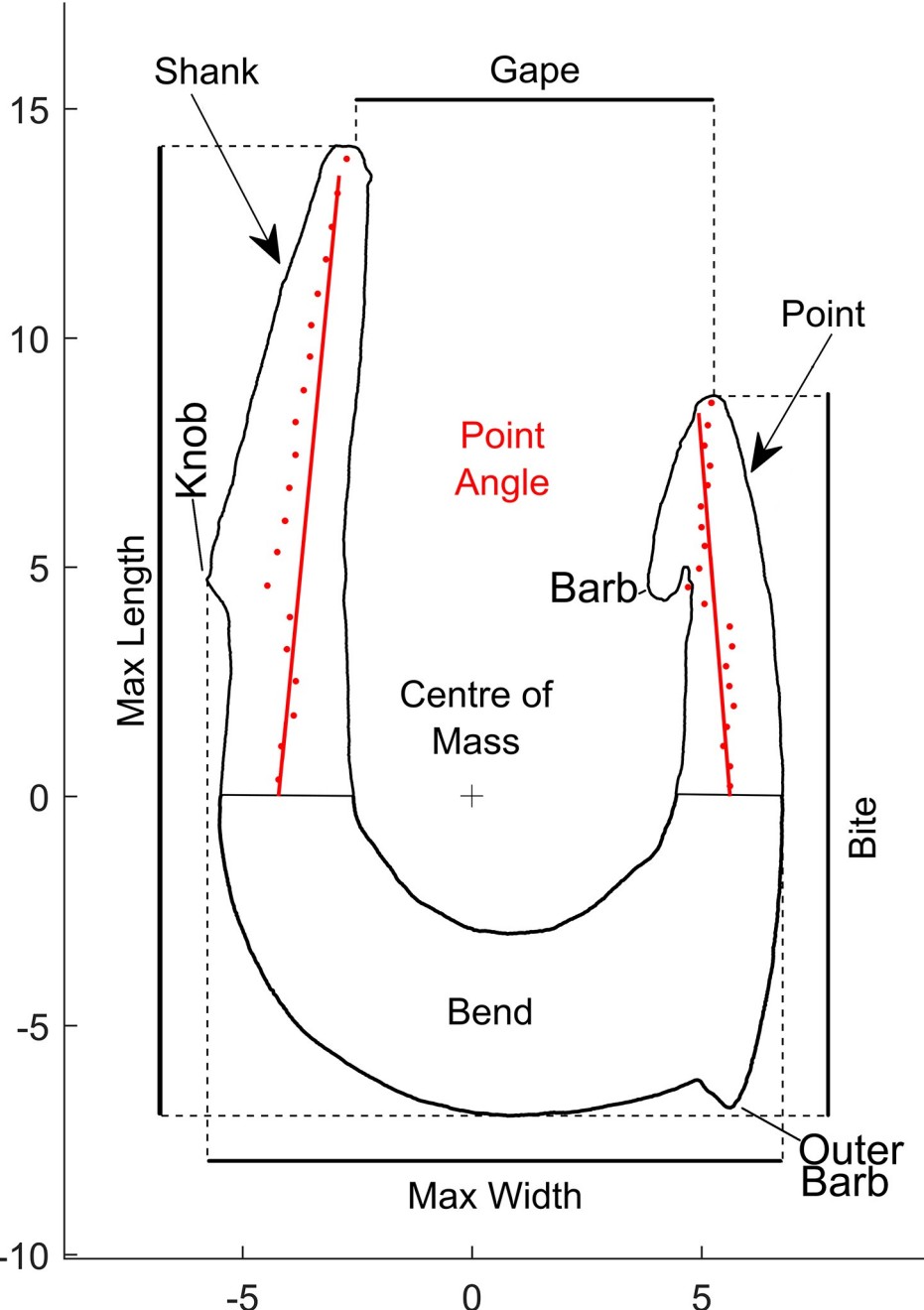

**Fig 5. Fish hook terminology and measurements used in the current study.**

introducing ambiguity in the classification and hampering the comparison among assemblages. In their study on Scandinavian fish hooks, Olson et al. [86] proposed a series of measurements of the linear dimensions and angles of the most common segments typically used to describe modern day fish hooks [87]. Given the importance of providing standardized methods of measurement to compare fish hooks from different times and places, and the challenge of taking accurate measurements of prehistoric hooks posed by their irregular shape and fragile nature, we extracted some of their measurements using high-resolution 3D scans. Digital

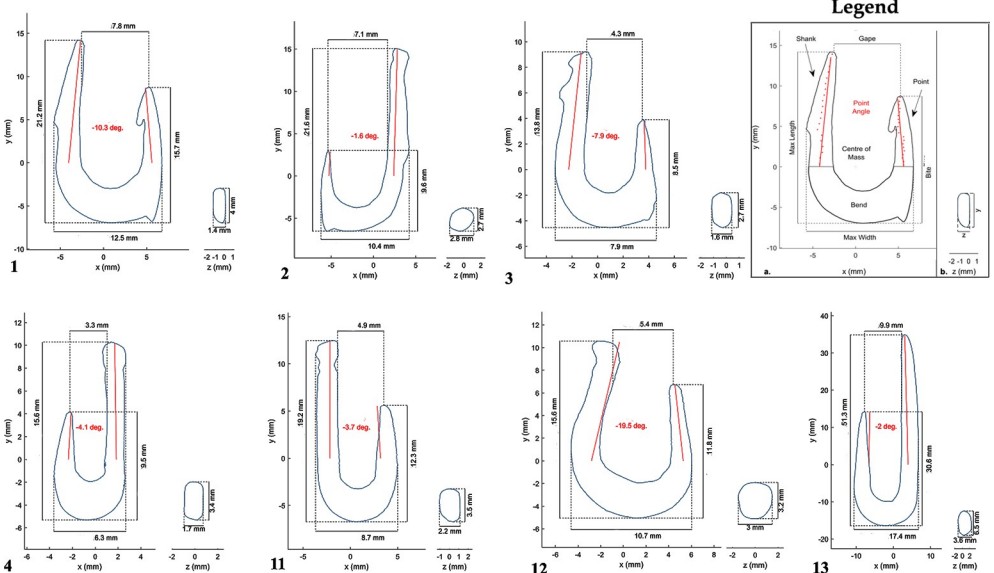

**Fig 6.** Measurements extracted based on the profile (legend a) and the cross-section (legend b) of the hooks.

3D models of the hooks were obtained using Polymetric® PTM-4c structured light scanners at the Computational Archaeology Laboratory (CAL) at the Hebrew University of Jerusalem. The use of high resolution 3D models allows precise and objective measurements to be taken and accurately replicated without having to handle these highly fragile artifacts, including parameters that cannot be measured using traditional methods such as the position of the center of mass [88]. In addition, the scans can be used with the open source *MeshLab* software [89] to virtually refit scanned fish hook fragments. Finally, the highly accurate 3D models provide a digital archive of these hooks, preserving them for future study.

The 3D models were manually positioned so that the hook's shank was parallel to the y-axis with the bend on the plane defined by the y- and x-axes using the *Artifact3-D* software [freely available upon request to CAL; 88]. Since JRD hooks are almost flat, a 2D outline was created for each complete or refitted item using the 'Process Object' function to simplify the measurement procedure. Outlines are defined by a series of vertexes with width and length coordinates. Based on the position of the center of mass (automatically defined by *Artifact 3-D*), the outlines were then segmented into three parts: the **bend** (below the x-axis), the **shank**, and the **point** (above the x-axis and on either side of the y-axis) (Figs 5 and 6).

We used the 3D models to extract the maximum length and width, gape, and bite (Fig 5). measurements proposed by Olson et al. [86]. In addition, the angle between the point and the shank was calculated. Finally, we used the 3D models to measure the width and thickness of the transversal cross-section of the bend at its lowest point and its maximum axis (width and thickness respectively) with the 'Create Cut' function in *Artifact3-D*.

The following measurements were taken from each hook when possible (Figs 5 and 6):

*Maximum length* (*L*; mm): distance on the y-axis between the lowest and highest vertex of the outline (length). Measured on all complete and refitted items.

*Maximum width* (*W*; mm): distance on the x-axis from the first vertex at the left and the last vertex on the right of the profile. Measured on all complete and refitted items.

*Hook size* (*HS*; mm): the square root of the product of bend thickness and height. Extracting the square root of the product allows comparison of values with different dimensionalities (1D

and 2D). Measured for all complete items.

$$HS = \sqrt{L \times W}$$

*Gape* (*G*; mm): distance on the x-axis between the highest vertex of the point and the shank.

*Bite* (*B*; mm): distance on the y-axis between the highest vertex of the point and the lowest vertex of the profile. Measured on all complete and refitted items.

*Point angle* (*PA*; deg.): angle between the point and the shank. The outline of the point and the shank was further subdivided into slices taken along the long axis (100 slices). For each slice, the mean of the length and width coordinates of all the vertexes was calculated. Two straight lines were fitted to best approximate the values of the mean coordinates of all the slices of the point and shank, respectively. The angle between the two fitted lines represents the angle between the point and the shank. Measured on all complete and refitted items.

*Bend thickness* (*T*; mm): distance on the z-axis between the most external vertices of the transversal cross-section of the bend at its lower point. Measured for all items.

*Bend height* (*H*; mm): distance on the x-axis between the lower and the higher vertices of the transversal cross-section of the hook bend at its lower point. Measured for all items.

*Bend size* (*BS*; mm): the square root of the product of bend thickness and height. Measured for all items.

$$BS = \sqrt{T \times H}$$

The measurement and the 3D digital models of the hooks are available in "S1 Table".

**Hook size results.** The JRD hook sample is too small to allow more than descriptive statistical analysis. The hooks vary considerably in all size dimensions and ratios (Fig 6, Table 2). Despite this variability, average maximum length of complete hooks declines over time (Fig 4; Table 2). The hooks from Layer 3b are larger on average than the hooks from Layer 3a, while hook #13, the only example from Layer D-8 (equivalent to Layer 3c), is the earliest and the largest of all. A similar trend is observed in the gape and bite dimensions and, although

**Table 2. Measurements of the JRD bone hooks by layer.**

| Context | | Length (mm) | Width (mm) | Gape (mm) | Bite (mm) | Angle (deg.) | Bend cs y (mm) | Bend cs z (mm) | Bend cs y/z | Bend size (mm) |
|---|---|---|---|---|---|---|---|---|---|---|
| Layer 3–0, D3 N = 4(1) | Avg. | 15.6 | 10.7 | 5.4 | 11.8 | -19.5 | 3.2 | 2.3 | 1.4 | 2.7 |
| | SD | | | | | | 0.3 | 0.4 | 0.3 | 0.3 |
| | Max | | | | | | 3.7 | 3.0 | 1.8 | 3.1 |
| | Min | | | | | | 2.8 | 1.9 | 1.1 | 2.3 |
| Layer 3a, D4 N = 7(2) | Avg. | 17.4 | 7.5 | 4.1 | 10.9 | -3.9 | 3.3 | 1.8 | 2.0 | 2.4 |
| | SD | 1.8 | 1.2 | 0.8 | 1.4 | 0.2 | 1.0 | 0.7 | 0.6 | 0.8 |
| | Max | 19.2 | 8.7 | 4.9 | 12.3 | -3.7 | 5.3 | 3.3 | 3.0 | 4.2 |
| | Min | 15.6 | 6.3 | 3.3 | 9.5 | -4.1 | 2.1 | 0.8 | 1.4 | 1.3 |
| Layer 3b, D6 N = 5(3) | Avg. | 18.8 | 10.3 | 6.4 | 11.2 | -6.6 | 3.4 | 2.1 | 1.8 | 2.6 |
| | SD | 3.6 | 1.9 | 1.5 | 3.2 | 3.7 | 1.0 | 0.7 | 0.6 | 0.7 |
| | Max | 21.6 | 12.5 | 7.8 | 15.7 | -1.6 | 5.1 | 2.9 | 2.8 | 3.9 |
| | Min | 13.8 | 7.9 | 4.3 | 8.5 | -10.3 | 2.3 | 1.4 | 1.0 | 1.9 |
| Total assemblage N = 18(7) | Avg. | 22.6 | 10.6 | 6.1 | 14.0 | -7.0 | 3.4 | 2.0 | 1.8 | 2.6 |
| | SD | 12.0 | 3.4 | 2.1 | 7.1 | 5.9 | 1.2 | 0.8 | 0.5 | 0.9 |
| | Max | 51.3 | 17.4 | 9.9 | 30.6 | -1.6 | 6.5 | 3.6 | 3.0 | 4.9 |
| | Min | 13.8 | 6.3 | 3.3 | 8.5 | -19.5 | 2.1 | 0.8 | 1.0 | 1.3 |

In the layer column N = total number of broken and complete hooks, the number of complete hooks per layer is given in parenthesis. mm = millimeters, deg = degrees.

weaker, in bend size. Finally, high variability in the point angle is observed both within and among the layers (Table 2).

## Morphological characters of fish hooks

Next, we describe each hook's overall shape. On the most robust level, we categorize the hooks according to the modern-day division of J-shaped and circular categories [76,90,91]. We then zoom in to characterize important, yet variable features of the JRD hooks including barbs that prevent the fish from escaping after it is captured and line attachment features. We do so by recording the form, location and presence or absence of barbs, grooves and knobs on the shank, and the point of the hook.

The most striking observation regarding the JRD hooks is their variability (Figs 3 and 4; Table 1). Variability is evident in all aspects of hook morphology–their size, angle and the presence and location of features and even the solutions for line attachment. The shape, size range and angles of the fish hooks fit well within the range of variation of modern fish hooks, but a few features found in the JRD hooks are not observed in present day hooks, nor in any early prehistoric hooks that we know of (see below).

In regard to their general morphology (Figs 3 and 4; Table 1), most of the complete JRD hooks belong to the J-type (#2, 3, 4, 11 and 13), while two hooks (#1 and 12) are more circular in shape, although not fully circular by modern criteria. Although no off-set hooks (where the plane of the shank deviates sideways from that of the point; [90] were found at JRD (or any Natufian site), the bended natural morphology of the long bones used as blanks for the production of hooks #1 and 3 may have served a similar purpose.

The most notable features on the JRD hooks include barbs and line attachment modalities (Table 1).

**Barbs.**   Four of the JRD hooks (#1, 2, 3 and 5) have barbs. Notably, all of these were recovered from Layer 3b. Only a single hook (#1) has an inner point barb, similar to the pointed barbs typical of the majority of contemporary hooks. The other three hooks (#1, 2 and 3; Figs 3 and 4; Table 2) have outer bite barbs (lower point barbs according to [78] located on the frontal part of the bend (point), just under the tip. Hook #1 has both an inner point and an outer bite barb, while the outer point barb on hook #5 is located at the middle of the point. The points of all other hooks at JRD are barbless. However, an interesting feature on three broken hooks from Layer 3a and its stratigraphic equivalent, Layer D4, may have served a similar purpose. In hooks #16 and 17, and to a lesser extent hook #15, the anterior part of the bend turns at an angle of close to 90 degrees, creating a sharp edge on the lower anterior bend that may have functioned like a lower point barb (Fig 4; Table 2).

**Line attachment features.**   Of the JRD hooks that preserve the shank, only the large, early hook #13 (Layer D8) lacks a line attachment feature on the upper half of the shank. All other hooks present a variety of line attachment solutions. A lone groove occurs around the circumference of the shank in two hooks (#2 and 4), while a knob is fashioned above a groove in two others (#11 and 12). Two other hooks lack a groove, but have a single knob located at the top (#1 and 6) or center of the shank (#2). Finally, one hook (#3) has a double knob at the top of the shank (Fig 4; Table 1).

## Functional analysis of bone fish hooks

**Methodology.**   The techno-functional properties of the fish hooks were analyzed at the DANTE- Diet and Ancient Technology Laboratory (Sapienza University of Rome) using a stereo-microscope ZEISS Axio-Zoom (10X to 165X) for low magnification observation and a ZEISS Axio Scope A1 (from 100X to 400X) for observation at high magnification. The study of

manufacturing and use traces as well as visual residue analysis (see below) were performed *in situ* to provide complementary data for understanding the modalities of hook function and use. Criteria selected for technological and use-wear identification were derived from the rich literature on osseous tool modification [92–99] and on bone use wear traces [94,98,100–102]. Specific studies presenting hook production traces include [103].

The primary goal of the analysis was to reconstruct the specific operational sequences related to the production, use and discard of the hooks. The type, intensity, location and distribution of technological use-wear traces, including flattening, rounding of the surfaces, striations, fractures, and residues, and the presence of transversal grooves on the distal part of the shank were recorded on each hook when present. Comparison was made with experimental fish hooks manufactured to resemble the JRD hooks and produced on bone using flint and basalt tools and with experimental hooks from the reference collection of DANTE laboratory, created for interpreting hook assemblages from the prehistoric Balkans [25,104]. Morpho-qualitative features of identified residues (e.g., color, appearance, inclusions, consistency, birefringence) were interpreted through direct comparison with the experimental residues from the same collection. These include a variety of residues used for hook hafting (e.g., natural, and ochre stained strings of hide and sinews, plant fibers, adhesive compounds such as bee-wax, resin, bitumen, animal glues). Only residues that show features such as patination, smearing, flattening, and directionality are considered reliable for functional interpretation. Published data have also been consulted to interpret the archaeological hooks [6,25,105–111]. In addition to the experimental reference collection, a selection of 30 ethnographic fishing hooks made of bone, antler and shell produced and used by Native American hunter-gatherers and New Zealand Maori foragers, served as references for the technological and functional interpretation of the archaeological artifacts. Ethnographic reference items from the Museum of Archaeology and Anthropology (MAA) in Cambridge (United Kingdom) and the Museum of Prehistory and Ethnography "Luigi Pigorini" were also used as references [25].

**Functional analysis results.** Reconstructing hook reduction sequences was challenging as no unfinished hooks, roughouts or manufacturing waste were identified at the site. Moreover, evidence left by intense sharpening and/or the prolonged use of the hooks had often worn off anatomical or technological features. Nevertheless, major trends could be identified. The technological analysis of the JRD hooks identified the following manufacturing phases (for hook parts terminology see Fig 5):

1. *Extraction of the blanks*: Rectangular blanks were extracted from the bone diaphysis through longitudinal and transversal cutting (Figs 7C, 7F, 8B, 8O and 9C) and roughly shaped through scraping and/or grinding (Figs 7F, 7I, 8B, 8O and 9C).

2. *Preparation of the bend*: drilling (probably using a flint? bit) was used to initiate the concave side of the bend of the hook (Figs 7B, 7G, 8I and 9F). When necessary, the area of the perforation was regularized by scraping or grinding (Figs 7K, 8I and 9F). It was not possible to determine whether the hook was perforated before or after the extraction of the blank as roughouts/manufacturing waste were not recovered at JRD.

3. *Creation of the shank/throat*: the shank, bend, point and the resulting throat of the hook were carefully regularized using longitudinal scraping. On the basis of our experiments, we can suggest that both flint and basalt tools were used, the first leaving very regular striations. Precise strokes were always applied along the shank (#4, 11, 6, 7, 14 and 13; Fig 4), the outer side of the bend, and the throat of the hooks. Sometimes scraping of the shank created a sinuous profile or left chatter marks, which are typical microscopic markers of this technique (Fig 7F and 7I).

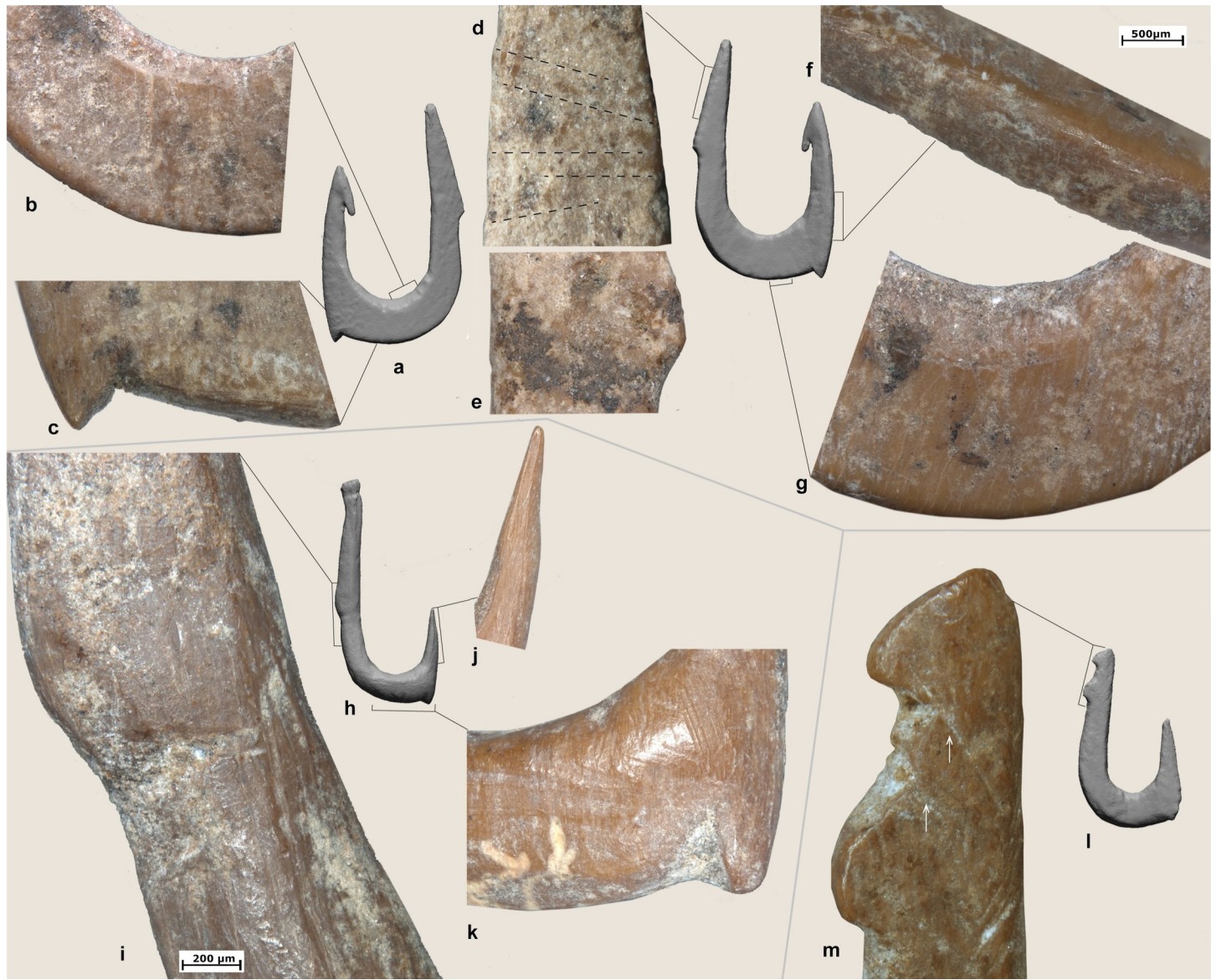

**Fig 7.** Technological traces and functional modifications on archaeological hooks (a) hook #1, h) hook #2, l) hook #3. b) Detail of the marks left during the preparation of the bend through drilling; c) detail of the barb and transversal grooving associated with its production; d) surface alteration associated with the decomposition of organic residues, still preserved on the other side of the hook; e) dashed lines indicate changes of color according to the disposition of the organic residues; f) scraping traces associated with the production of the shaft; g) perforation marks; i) j) k) transversal grooving applied to produce the barb; m) detail of the top of the shank. The arrows indicate technological mistakes produced by a lithic tool for creating a double knob for line attachment.

4. *Creation of the point/barbs*: Points were sharpened through regular scraping (Fig 7J). In a few cases, barbs were created through lateral transversal grooving (#1, 2, 3 and 5; Fig 7C and 7K).

5. *Line attachment features on the shank*. Most of the features on the hooks were created by grooving. Grooves were created around the full circumference of the distal end of the shank of four hooks (#2, 4, 6 and 12) to create a small globular head (Fig 8K and 8O) or in three cases, on the lateral side to create single or double notches (e.g., #1 and 3 and 11; Figs 7M

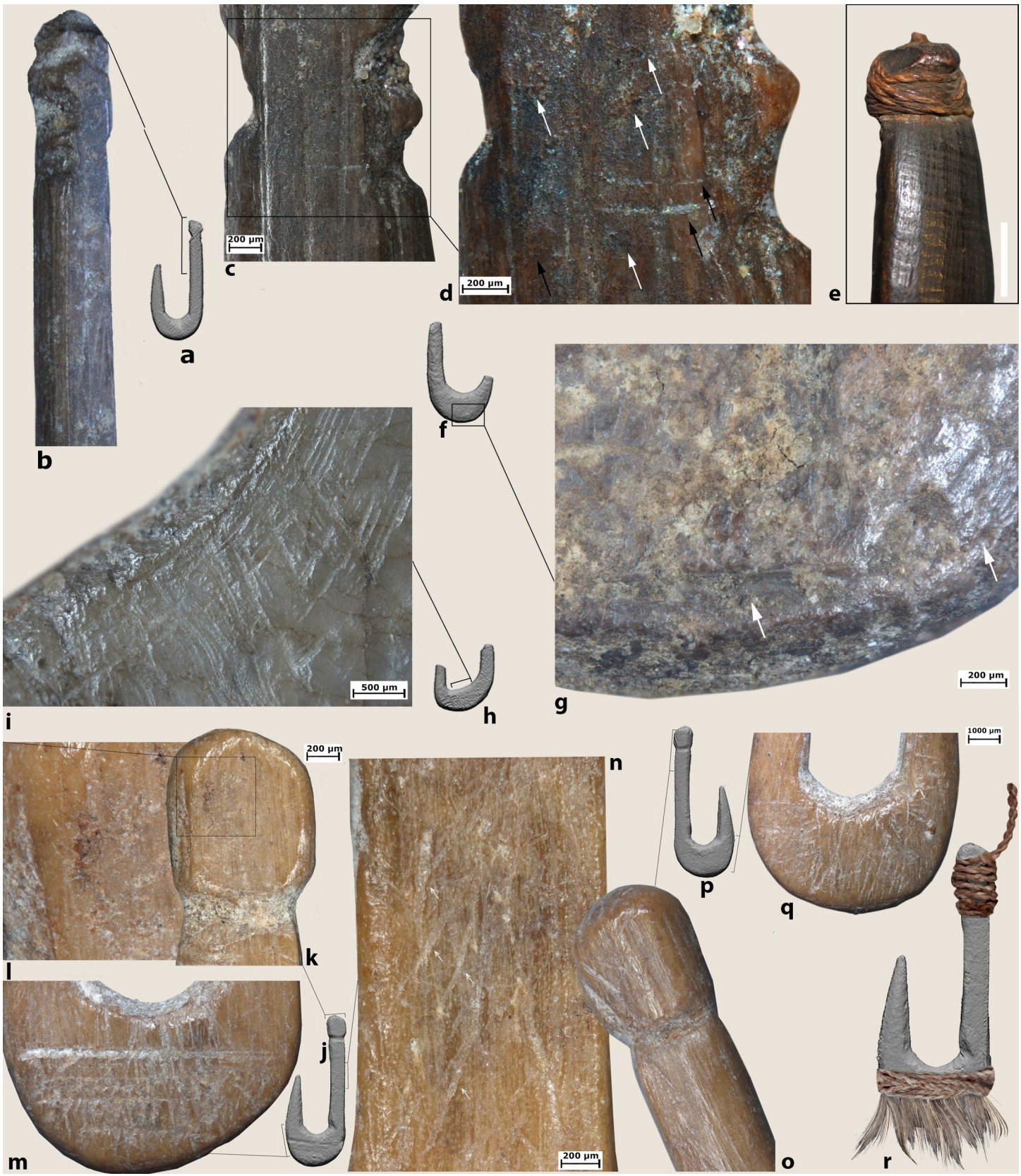

**Fig 8.** Technological traces, use wear traces and residues on archaeological hooks (a) hook #14; f) hook #16, h) hook #16; j,p) hook #4). b) longitudinal cutting marks produced when extracting the rectangular blank from the bone diaphysis; c, d) details of functional features (rounding, striations and residues) related to the attachment of the line; e) close-up photos of the line attachment system of an ethnographic lure hook with details and distribution of use traces and residues (hook # 25.445) from the Museum of Archaeology and Anthropology (UK); g) transversal cutting marks; i) marks associated with drilling used to create the concave side of the bend of the hook; k) rounding observed on the knob and particles of residual brownish adhesive matter inside the notch produced for creating the knob; l) particles of residual brownish adhesive matter; m) deep transversal incisions across the bend of the hook likely used to connect a cord for a lure. Note patches of residual brownish adhesive matter; n) short oblique and superimposed striations characterized by an initial depression and rough bottom, possibly fish tooth marks; o) rounding on the knob. Note also the longitudinal cutting marks left during the preparation of the hook; r) functional reconstruction of the hook according to archaeological use wear and residues.

and 8B–8D). In one case, a lateral knob was also part of the hook's line attachment system (Fig 7I), while the shank was worked into a point on another (Fig 9A).

6. *Technological adjustments on the bend*. In two cases, deep transversal incisions were made across the bend of the hooks using a stone tool (#4 and 9). These likely served to connect a cord for a lure (see below; Figs 8M and 10D–10F).

All of the JRD bone fish hooks show wear traces related to prolonged use as well as residues connected to line attachment features and when present the use of lures. At low magnification, rounding is observed on the hook's shanks and especially inside the notches created by the knobs (Figs 7M and 8C). Use-traces are also visible across the knobs (Figs 8B, 9C and 9D), inside the shank grooves (Fig 7I), across the bends (Figs 7K, 8D and 11D), along the upper part of the shanks, and on the tips (Fig 10A). Fine striations, transversally oriented to the main axis of the hook located on the shanks (Fig 9C and 9D) and line attachment features (Fig 7D, 7E and 7I) relate to the connection and securing of the line.

At high magnification patches of bright polish with flat topography and fine transversal striations have been identified on the upper parts of the shank (Fig 9B). On hooks #1, 4, 9, 13 and 14, rounding and striations on the line-attachment features are associated with residues of clear feature and orientation (Figs 7D, 7E, 8C, 8D, 8K, 8L, 9D, 9E, 10B and 10C). In particular, three types of residues have been identified: (a) particles of yellowish/brownish adhesive matter, rich in charcoal inclusions, and characterized by a flattened profile and transversal impressions (Figs 8C, 8D, 8L, 9D and 9E); (b) plant fibers (Fig 10B and 10D); and (c) animal hair (Fig 10E and 10F). Wear traces and yellowish/brownish residues are remarkably well preserved on the shank of hook #13, suggesting that adhesive was applied to the fibers to better secure the hook to the line (Fig 9D and 9E). Hooks #3, 4, 9 and 11 bear similar traces of line attachment method and residues (Figs 8C, 8D, 8K, 8L, 10B and 10C). A plant fiber was found inside the notches that form the line-attachment of hook #9 (Fig 10B and 10C). On hook #1, transversally oriented post-depositional bacterial modifications of the surface of the shank are associated with spots of brownish residues suggesting the coiling of threads around the shank (Fig 7D and 7E).

Interestingly, a few particles of adhesive and one long hair fiber were identified on the bends of hook #4 and 9 respectively (Figs 8M, 8Q and 10D–10F). The combination of such residues and evidence for the technological adjustment of the bend through deep transversal incisions hints at the use of lures at JRD (Fig 8R).

Use-wear traces are less developed on the points and the barbs than on other hook parts probably due to resharpening of the hook point after prolonged use. Along with traces and residues produced by prolonged use, the shaft of hook #9 is marked by short oblique/transversal striations, sometimes superimposed, and characterized by an initial depression, rough bottom and irregular walls. These traces resemble fish tooth marks such as the ones documented on experimental as well as ethnographic hooks (Fig 8E; [25,112]).

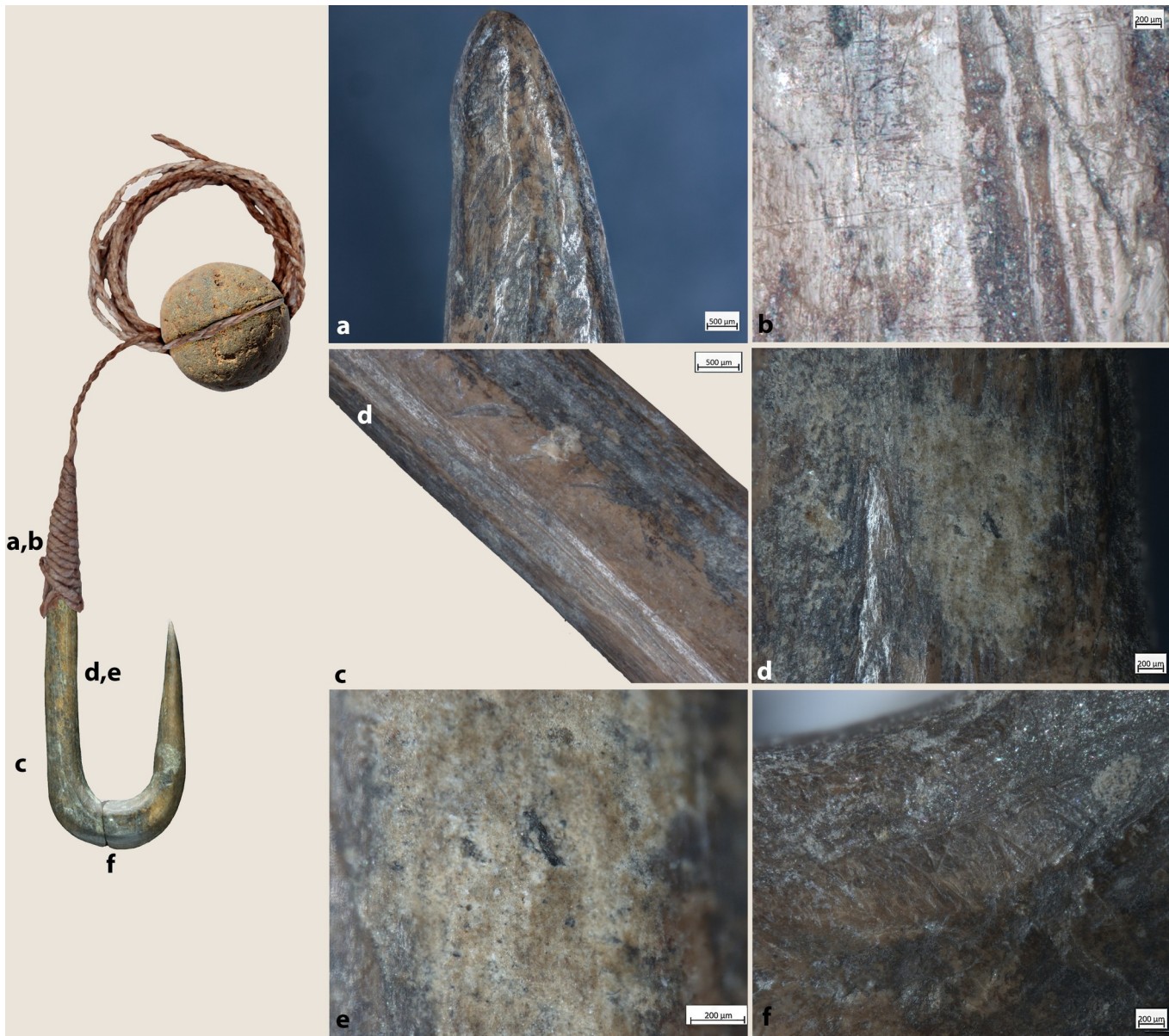

**Fig 9. Technological traces, use wear traces and residues on archaeological hook #13.** a) functional reconstruction of the hook according to archaeological use wear and residues; a) rounding developed on the tip of the shaft; b) bright polish with flat topography and fine transversal striations identified on the upper part of the shank; c) longitudinal cutting marks produced to extract rectangular blanks from the bone diaphysis; d,e) particles of residual brownish adhesive matter with clear transversal features; f) marks associated with drilling used to create the concave side of the bend of the hook and subsequent abrasion.

## Small grooved pebbles

### Materials

Ten small, grooved pebbles were recovered at JRD. These pebbles are less than 3 cm in maximal diameter and are bisected by a single or, in one case, double groove (Table 3; Fig 11).

Six of the ten grooved stones are produced on smooth basalt pebbles and four on limestone. The mean maximal dimension of the pebbles is 17.4 mm (s.d. 4.5 mm), the width is 13.4 mm on average (s.d. 4.1 mm) and the mean circumference is 52.0 mm (s.d. 14.0). The average weight is 4.8g with a very large s.d. of 4.2g reflecting a wide weight range from 1 to 14.5g.

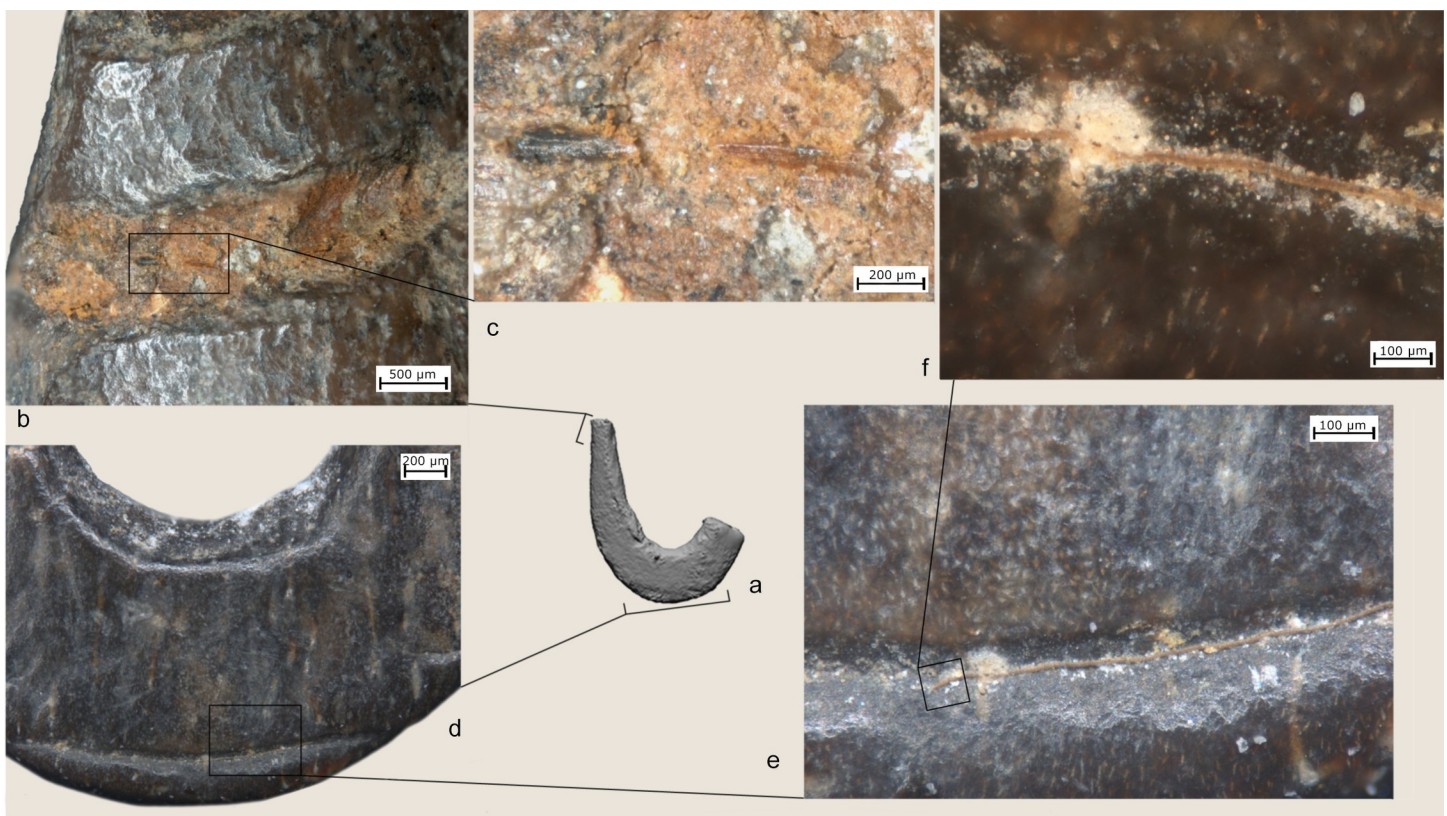

**Fig 10.** Technological traces, use wear traces and residues on archaeological hook #9 a,b,c) details of a plant fiber identified inside the notch on the top of the shank; d-f) animal hair recovered inside the groove on the bend of the hook.

Sample size is not large enough to allow statistical analysis, but the distribution shows a concentration of pebbles around 5g in weight with both small and large outliers. The grooves run either longitudinally, or more commonly, transversally to the main axis of the object (Fig 11), and one case shows a double parallel groove (pebble #110; Fig 11).

Six of the ten grooved pebbles found at JRD were analyzed in detail for the current study (Fig 11; Table 3). Two of the analyzed pebbles are basalt and four are limestone.

## Archaeological context

Of the ten grooved pebbles from JRD, four were identified during excavation and six were found during sediment sorting. Six pebbles originated from Layer 3a, the latest Natufian layer at JRD (Fig 2). Two pebbles were obtained from Layer 3c and a single pebble is from the lower Natufian Layer D6 in Area D (equivalent to 3b). Thus, 90% of the grooved pebbles originate from the Natufian layers of the site. The only exception is pebble #106, which was found in Layer 5. The groove around pebble #106 is not as deep as the grooves on the other pebbles and it may represent an unfinished product (Fig 12A). Importantly, with the exception of pebble #106, the archaeological context of the small grooved pebbles matches that of the bone fish hooks.

## Functional analysis of the small grooved pebbles

**Methodology.** The artifacts were first inspected at the DANTE Laboratory (Sapienza University, Rome) and subsequently analyzed at the TraCEr Laboratory (RGZM-MONREPOS,

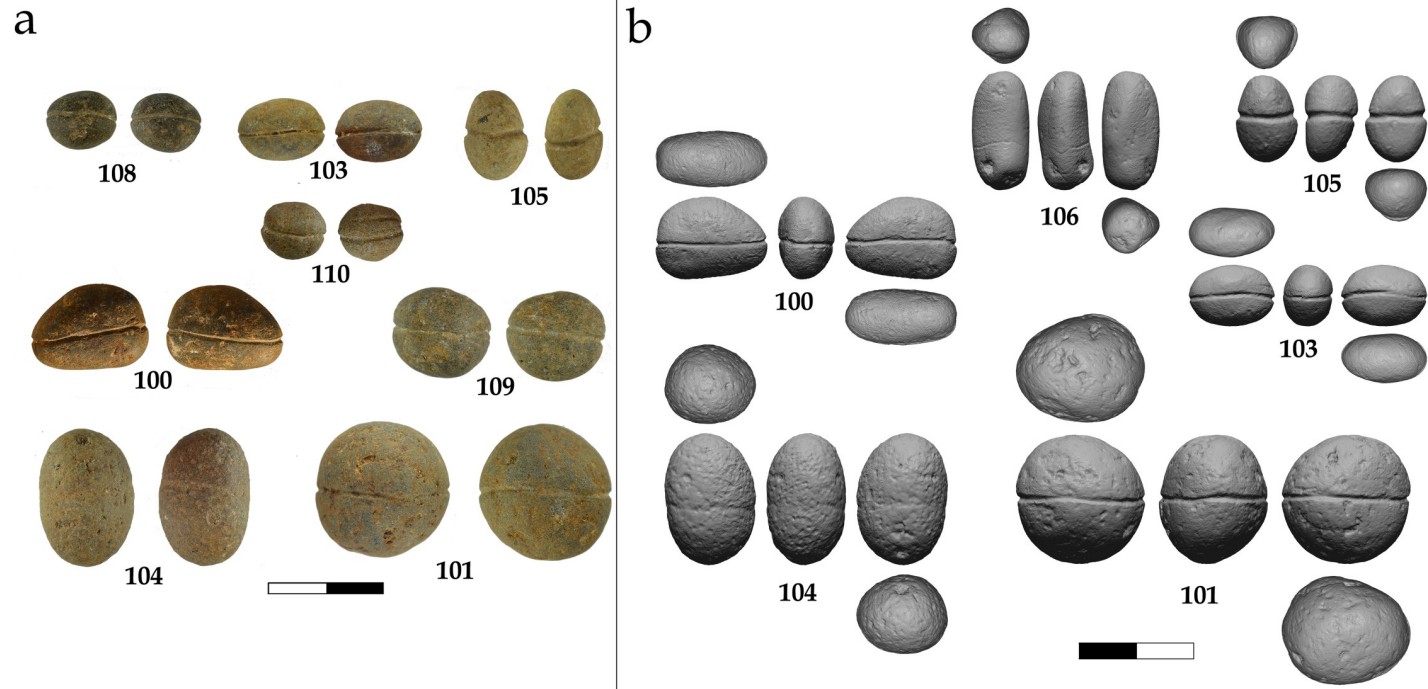

**Fig 11. Grooved pebbles from JRD.** a) Selected grooved pebbles; b) 3D models of the six analyzed pebbles. Numbers are according to site data catalog.

Germany). Two stereomicroscopes (a ZEISS Axio-Zoom V16 and a ZEISS SteREO Discovery V8) and two metallographic microscopes (ZEISS-Axio ScopeA1) were used for the study. The magnification ranged from 10x to 168x for observations with the stereomicroscopes, and from 50x to 200x with the metallographic microscope. Extended Depth of Focus (EDF) images were created using the post-processing software Helicon Focus.

During the first stage of analysis, artifacts were not cleaned beyond removing the sediments by hand to ensure preservation of any *in situ* residues. After making the first detailed observations, residue samples were collected [113,114] using a pipette and distilled water as a solvent.

**Table 3. Size measurements of the JRD grooved pebbles.** The JRD grooved pebbles are reposited at the Prehistory Laboratory, Tel Hai College, Israel. The 3D grooved pebble models are available (wrl format) at https://doi.org/10.7910/DVN/VTKRQW.

| ID # | Area | Layer | Level top | Square | Raw Material | Weight (g) | Max length (mm) | Max width (mm) | Circum. (mm) | Remarks |
|---|---|---|---|---|---|---|---|---|---|---|
| 100 | B | 3(a?) | 57.22 | O100 | Limestone | 4.6 | 19.7 | 14.5 | 54 | Analyzed |
| 101 | B | 3a | 57.235 | O103 | Basalt | 14.5 | 23.4 | 23 | 78 | Analyzed |
| 103 | B | 3a | 57.336 | P103 | Limestone | 2.2 | 15.2 | 10.9 | 45 | Analyzed |
| 105 | B | 3c | 57.03 | Q100 | Limestone | 2.1 | 15.6 | 10.8 | 47 | Analyzed |
| 104 | B | 3c | 56.5 | M98 | Basalt | 8 | 23.4 | 15.6 | 67 | Analyzed |
| 106 | B | 5 | 55.982 | O99 | Limestone | - | 21 | 10 | - | Analyzed |
| 107 | B | 3a | | O105 | Limestone | 4.3 | 16 | 14.5 | 53 | Not analyzed |
| 108 | D | D6 | 57.38 | Q122 | Limestone | 1.8 | 12.3 | 10.3 | 37 | Not analyzed |
| 109 | B | 3a | | R102 | Limestone | 4.7 | 17.1 | 15.3 | 54 | Not analyzed |
| 110 | B | 3a | 57.4 | Q105 | Limestone | 1 | 10.2 | 9.2 | 33 | Not analyzed. Double groove |

g = grams, mm = millimeters.

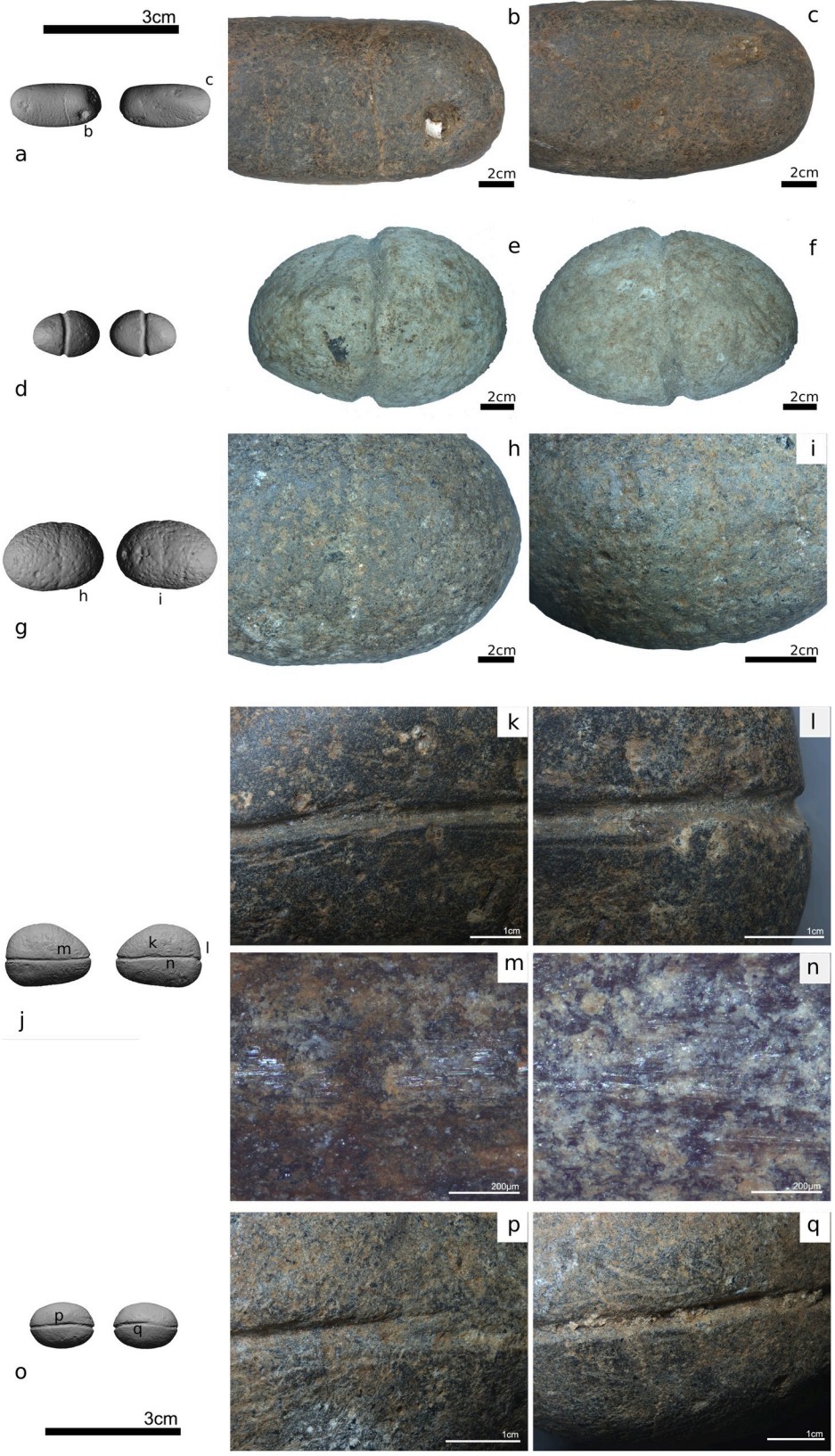

**Fig 12.** Technological modifications (grooves with variables depths) of a) pebble #106, d) pebble #105, g) pebble #104. Details of the incised grooves with rare technological errors indicated as shallow lines parallel to the main groove. i-k) Microscopic features on the surface of pebble #100. The groove is particularly deep (i, k) and technological errors are evident. Linear polish was also observed within the groove (m, n). f-h) Microscopic features on the surface of pebble #103. o,q) Details of the groove of pebble #103.

A total of four extractions using 100μl of distilled water were performed (three on pebble #103 and one on pebble #100; S1 Fig). Residues were identified by comparing them to published images [113,115–118]. The extracted residues were analyzed under transmitted light microscopy at TraCEr (ZEISS Axio Lab. A1). After sampling was completed, the archaeological tools were cleaned with water to expose their surfaces for analysis. Both technological marks of manufacturing and use-wear traces were recorded and described. When present, post-depositional surface modifications were also described.

**Results.** Other than limestone pebble #105 which was chemically altered by post-depositional erosion (Fig 12E and 12F), the surfaces of the small grooved pebbles were well preserved (Table 3). As is typical, the identification of use-wear traces on natural, unmodified stone surfaces was challenging [119].

Technological traces are divided into tool preparation marks (e.g., incision of the groove) and manufacturing errors. Grooves vary in depth and follow the longitudinal (pebbles #100, 101 and 103) or vertical axis (pebbles #104, 105 and 106; Fig 12). Variation in the depth of grooves on the superior and inferior faces is sometimes observed on the same implement (Fig 12B, 12C, 12H and 12I). The depth of the groove is not correlated with the raw material, as a range of depths values were observed on both limestone and basalt. The grooves were produced by the sharp edges of stone tools, as suggested by their irregular outlines and the variation in depth (Fig 12E, 12F, 12K and 12L). Technological errors, such as narrow, shallow micro-grooves oriented parallel to the main groove, are common (Fig 12K, 12L, 12P and 12Q). These were likely created when the stone tool slid out of the main groove during manufacture.

The studied pebbles show limited use-wear traces. This may be related to their short duration of use and/or the nature of their use. For example, if the small grooved pebbles were used only briefly as weights connected to a line of soft material, then only minimal traces are expected. In addition, one limestone specimen was subjected to post-depositional chemical alteration of its surface, making observation challenging (Fig 12A).

The only wear connected to use are lines of polish located inside and parallel to the groove (Figs 12M, 12N and 13H) on two of the basalt pebbles (#100 and 106; Figs 12J and 13F). No polish was observed on the limestone specimens, which are also slightly smaller than the basalt pebbles This pattern may be related to the nature of the raw material itself. Another study [120] has shown that polish forms much more slowly on limestone than on chert and usually presents a much less shiny surface.

**Results of residue analysis.** Residues, most with an environmental origin, were detected on all analyzed implements (n = 6). Undefined dark spots were observed on limestone pebble #105 (Fig 13A). Although no chemical analyses were performed to determine the nature of this residue, it likely derived from the depositional environment at the site. The visual appearance of the residue suggests that it was a mineral oxide. Fragments of translucent mother-of-pearl (Fig 13B), or smashed shell powder combined with sediment particles (Fig 13C and 13D) were found on all artifacts. These derive from the lakeshore sediments, which were extremely rich in mollusk shells in most layers [4]. Micro-fragments of shells were also found on all samples analyzed with transmitted light microscopy (S2A Fig) as well as sediment particles that were also sometimes visible (S2C and S2D Fig). Raphides (S2B Fig) were observed on one sample taken from pebble #103 (extraction location shown in S1E Fig). Interestingly, two degraded

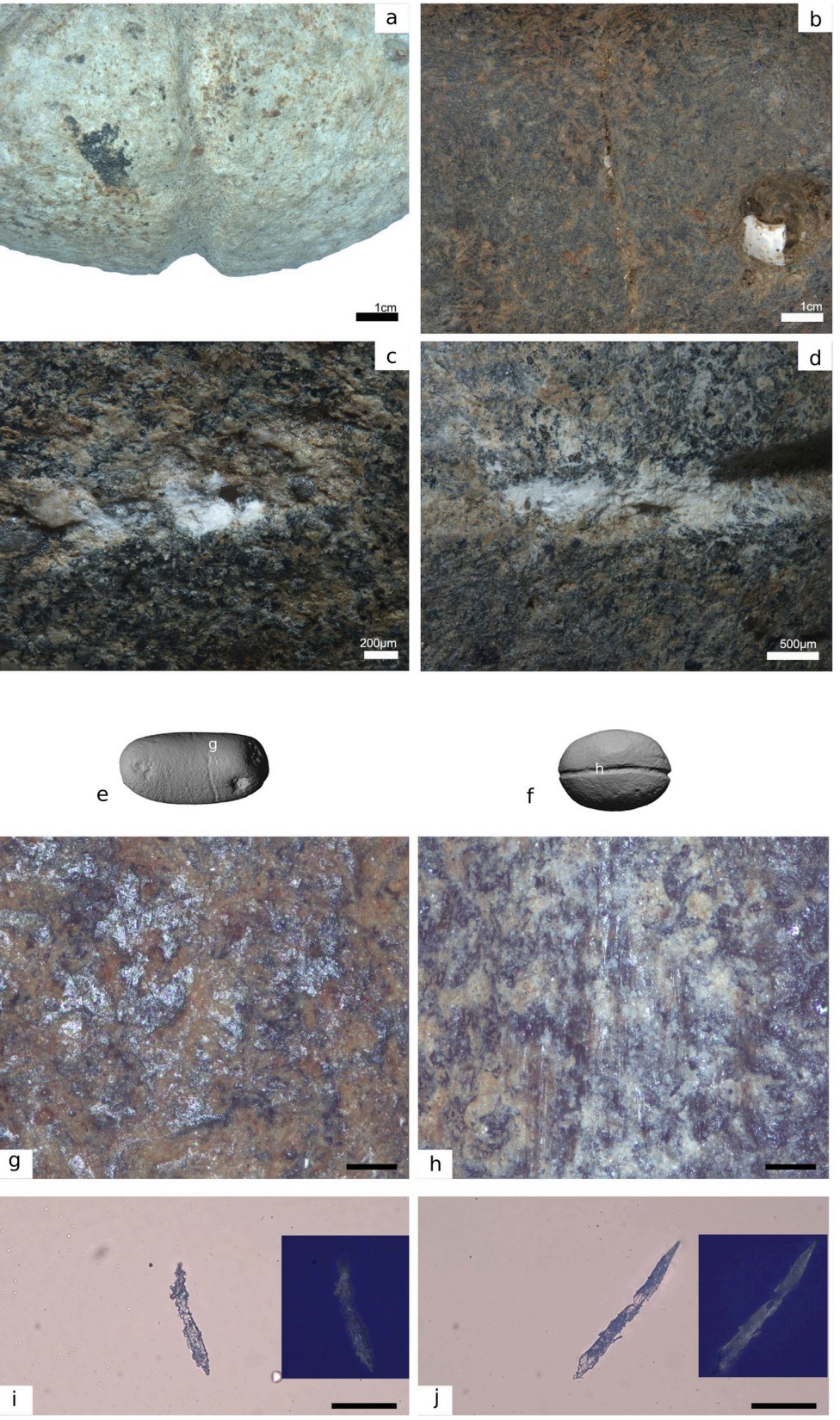

**Fig 13. Some residues observed on small pebbles #103, 105 and 106.** a) dark spots, possibly oxides, on pebble #105; b) a shell fragment on pebble #106; c-d) smashed shell powder mixed with sediment on pebble #103; e) pebble #106; f) pebble #103; g) polished area with randomly oriented striations; h) linear bands of polish located in the interior of the groove; i-j) fragments of vegetal fibers under plane polarized and crossed polarized light from pebble #103. Scale bars measure 100 μm.

fragments of vegetal fibers were also identified (pebble #103, Fig 13I and 13J) from the interior of the groove (S1D Fig) of the same artifact. These fibers could be remnants of the line used to tie the grooved pebble to fishing equipment.

## Fish remains

### Methods

The preliminary zooarchaeological data presented here includes the identified fauna from the 2016 excavation season (NISP = 413). The 2016 assemblage includes fauna from Layers 3–0, 3a, 3b, 3c, 4 and 5 and thus the samples for each individual layer are small (NISP = 11, 145, 49, 15 119, and 74 respectively). Although the hooks and grooved stones are found only in the Natufian Layers 3a 3b & 3c, we present data from all of the site's layers to highlight differences in the importance and composition of the assemblage that may be related to technological change.

This report focuses exclusively on the fish remains. Importantly, although the fish bones from the sample have been recorded in the database, only general preliminary identifications have been made to date. The three fish families identified thus far (*Cyprinidae*, *Cichlidae* and probable *Salmonidae*) can be distinguished based on the morphology of their vertebrae, teeth and head elements. The identification of species within these families is more challenging and only possible for certain elements (e.g., axis and atlas, teeth). Although most vertebrae cannot be identified beyond the family level, the maximum diameter of the centrum provides a rough proxy for fish body-size, that can further narrow down the potential species. For example, using modern fish referents Zohar et. al. [49] determined that Cyprinid vertebrae centra with diameters wider than 3.5mm can belong to only to fish great than 220 mm in length and thus must belong to one of three large Cyprinid species (*Luciobarbus longiceps*, *Casiobarbus canis*, *Capoeta damascina*) inhabiting the Upper Jordan Valley.

### Results

Because it was retrieved from water-logged deposits, the JRD fauna is in an excellent state of preservation. The sample includes fauna recovered from 2 mm mesh that was later washed and picked to recover small and delicate elements including tiny rodent and fish bones. The JRD fauna is comprised of diverse terrestrial and aquatic species represented by ungulates, carnivores and small game animals such as snakes, tortoises, turtles, birds, hares and fish. The ungulates that dominate Natufian assemblages at other sites are unusually rare at JRD, especially in Layer 3–0, 3a, 3b and 3c (<7%). Instead, in terms of raw frequency counts (number of identifiable specimens-NISP), fish is the most abundant taxon in every layer at the site. Nevertheless, there are fundamental differences in the relative abundance of fish across the layers, with the assemblages from Layers 3–0, 3a and 3b comprised of more than 80% fish, compared to only 27% fish in Layer 4 and more intermediate frequencies of fish in Layer 3c (55%) and 5 (45%; Fig 14A). Thus far, members of the Cyprinid (vertebrae, teeth and pharyngeal bones) and Cichlid families have been positively identified in the 2016 assemblage. Cyprinids are by far the most common family in all layers and include both large and small individuals. Teeth from the Cyprinid species *Luciobarbus longiceps*, *Casiobarbus canis* and *Acanthobrama*

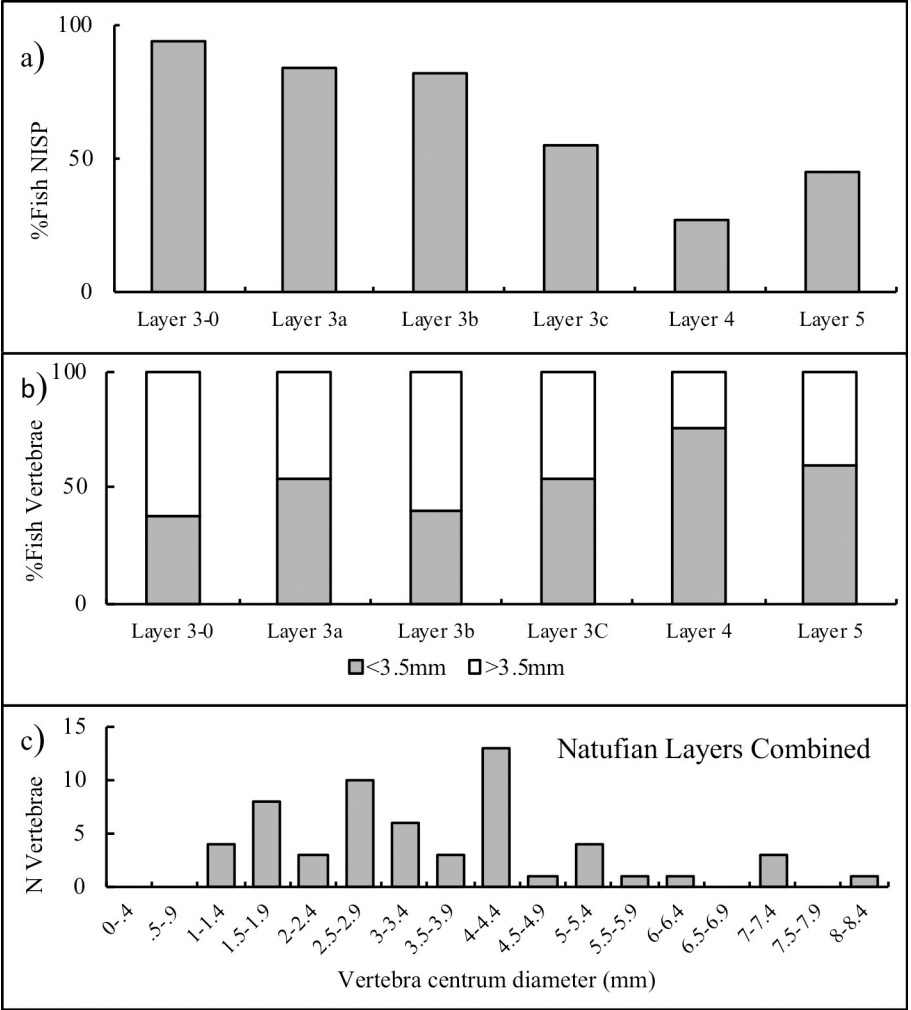

**Fig 14.** a) Relative abundance of fish at JRD by arcaheological layer; b) The proportion of fish with vertebral centrum diameters greater and less than 3.5 mm. c) Frequency distribution of fish vertebral centra diameters in the combined Natufian layers (3a, 3b, 3c).

*hulensis* have been identified thus far. A few Salmonids (e.g., trout) may also be present (see also [121]), but these identifications must be confirmed with an appropriate comparative assemblage.

The fish from the Natufian layers are larger in size than those from Layers 4 and 5. Between 45–65% of the fish vertebrae from Layers 3–0, 3a, 3b and 3c (52% for the combined Natufian layers) at JRD are larger than 3.5 mm in diameter, while only 25% in Layer 4 and 40% in Layer 5, are larger than 3.5 mm in diameter, (note that the sample sizes for the latter two layers are very small; Fig 14B). The combined distribution of vertebral centra diameters for the Natufian layers (3a, 3b, 3c) shows that they derive from fish with wide-ranging body-sizes (Fig 14C).

## Discussion

Although older fish hooks made of shell have been found in Southeast Asia and bone hooks appear at contemporaneous Natufian sites in the Levant, the JRD fish hook assemblage is unprecedented for its size up to this point in prehistory. Not only this, but the assemblage

preserves the earliest examples of significant technological advancements including barbs and the use of lures (but see slightly earlier dates for a possible shell lure from Makpan Cave [122]. The association of this important assemblage with a collection of grooved stones interpreted as line weights and well-preserved fish remains enables us to discuss not only the hooks, but the fishing technology and techniques, and the larger practice of fishing at JRD and its evolution over time. The rich reconstruction afforded by this study was made possible only through our multidimensional analytical approach that combined archaeological, technological, micro-scopic use wear, residue and zooarchaeological lines of evidence.

Importantly, all of the JRD bone fish hooks and small grooved pebbles (line weights) originate from the Natufian layers and all but one are found in clear stratigraphic context. The hooks are widely distributed across the site (Table 1)—only a single 1x1 m excavation unit (Square O102), yielded more than one hook. This distribution is explained by the nature of the activity and the accumulation environment. This pattern of deposition reveals that hooks were used for fishing in shallow water close to the lakeshore and were likely deposited in this nearshore environment when they were lost or broken during use. That the deposits represent an active fishing rather than a preparation area is supported by the absence of hook manufacturing waste at JRD. This dif-fers from habitation sites such as Natufian Eynan [57] or Mesolithic and Neolithic sites in north-ern and eastern Europe [82,86,123,124] where manufacturing debris is present. The sedimentological evidence from JRD also indicates that the site deposits accumulated in a near-shore environment [4]. Hook loss is a regular occurrence during fishing and would have resulted in the sporadic accumulation of fish hooks at the bottom of the lake bed.

### Features of the fish hooks

**Barbs.** Although barbed points are widespread in Europe by the onset of the Last Glacial Maximum, the barbed fish hooks from JRD are, as far as we know, the earliest examples identi-fied to date and the only ones found in Natufian sites. As with bone points, barbs prevent prey from freeing themselves from the hook increasing the odds of successful capture. More than one-third of the hooks at JRD have barbs, and all are found in Layer 3b. Present day fishers suggest that inner point barbs like the one on JRD hook #1 function to secure the bait to the hook rather than to prevent the escape of the fish and some barbed hook types are actually named "baitholders" [80]. All other barbs at JRD are located on the outer side of the frontal bend of the hook. In other cases, the bend and the point are oriented at $90^0$ to one another, cre-ating a morphology that resembles an outer barb and may have functioned the same way. Outer barbs (out-barbs or obverse barbs) are rare in modern metal fish hooks and in prehis-toric assemblages but appear in some Pacific, especially New Zealand and Hawaiian sites [78]. Their function has been described both as an anchor to attach bait lines [125] or as a less risky alternative to the internal barb which often tangles or damages lines when they go slack (fish hook manufacturer Gamakatsu®).

The JRD barbed hook sample is small, yet the presence of both inner and outer barbs and the variety of morphologies and barb locations show that even though these are the earliest examples of barbed hooks, the Natufian anglers were familiar with the full range of their func-tions. The barbed hooks appear only in Layer 3b, the second of three Natufian layers at the site, after fish hooks had already been in use at the site for at least 1000 years [4], thus evidenc-ing the evolution of hook fishing technology over the site's use. Interestingly, barbs are also absent from hooks in later levels at JRD and at other Natufian and Neolithic sites in the south-ern Levant [66].

**Line attachment.** The residue analysis and plant fibers recovered from the shaft of the hooks and the grooved pebble weights indicate that the fishing lines were made of vegetal

material. The identification of plant species used for the production of the fishing line is particularly challenging, especially if fibers are observed under reflected light. The rich botanical assemblage from JRD includes both trees (such as willow) and water plant species like bulrush or cattail [4,69] that are suitable for producing fine and resistant cordage [126]. Judging from the size of the hooks and their grooves, the presence of the small grooved pebbles, and the fish taxa captured at the site, the lines used for the attachment were likely strong enough to pull a 1kg and possibly even heavier fish out of the water [126].

A secure and strong connection between the line and the hook is one of the most important requirements of line fishing. The risk of losing a hook or line weight if the line loosens when a fish is caught or the hook becomes entangled with stones or plants is well known to anglers. The method used to tie lines to the hooks and weights can be reconstructed from line attachment features on the hooks, as well as use wear traces and residues identified on the hook shafts. Like the other features described thus far, the JRD bone hooks and the small grooved pebbles provide evidence for complex and diverse line attachment methods. The hooks from JRD or other prehistoric sites do not have eyes or holes in the shank to thread the line, perhaps because it weakens the narrow shank. Instead, a variety of other solutions to ensure a strong line connection are documented at JRD. These include grooves, single or double knobs on the hook shafts or a combination thereof (Table 2). The wear traces left on the shank of the hook by the lines (Fig 7M, 8C and 8D), and the location of the knobs on some hooks (e.g., hook #1) indicate that the line connection covered a large part of the shank. The line was not connected by a single twist around the shank but by a complex method of binding, wrapping and tying [25]. Furthermore, the results of the residue analysis indicate that an adhesive was also used to secure the hook to the line.

**Lures.**   The presence of grooved lines at the lower part of the bend in hooks # 4 and #9 in addition to the use wear and residues observed on the hooks from Layers 3a and 3b, confirm the early use of artificial baits (lures) at JRD. Through their colors, movements and vibrations under water, lures typically mimic invertebrates, and baitfish, attracting a fish's attention. Today, artificial light weighted baits (artificial flies) are commonly used to catch carps and members of the Salmonidae family, such as salmon and trout. The use of lures by prehistoric anglers reflects detailed knowledge of fish behavior and diet.

Although lures are well known ethnographically and in modern fishing, they are difficult to identify archaeologically. Thus far, the use of artificial light-weight baits attached to hooks has been identified through detailed microscopic analysis only at the Late Neolithic site of Vinča–Belo Brdo in Serbia [25]. Just recently, three small shell fragments from the site of Makpan Cave in Indonesia have been as interpreted possible lures [122]. One of these originates from a context that dates to about 13,600 cal BP. At JRD, the presence of deep grooves, adhesive, and animal hair on the bend of two hooks indicate that lures were already used at the end of the Natufian, making the JRD lures one of the oldest, if not the oldest, yet discovered. At JRD, artificial baits may have included shell flutters–flat plaquettes of shiny mother-of-pearl that wiggle in the water to attract fish (also see [82] for possible bone "flutters" in Mesolithic Norway). The JRD Natufian horizons are rich with large *Unio* bivalve shells, suitable for shell flutters. Shell flutters and shell lure shanks are ethnographically well known in the Pacific for their ability to attract fish. The mother-of-pearl "effect" is also exploited in modern fishing industry to produce artificial flies. Today, the use of light lures requires a specific casting technique, known as fly fishing. Given the small dimensions of the hooks likely to have been equipped with artificial lures at JRD, the possibility that a similar angling method was already in use during the Natufian should not be ruled out.

## Fish hook assemblage diversity

Perhaps the most important aspect of the JRD fish hook assemblage is its substantial variability which is even more significant given the very old age of the assemblage. There are no two similar hooks at the site. Hooks vary in size, style, morphology, features and line attachment methods (Table 1) and this variation occurs both within and between the archaeological horizons. The process of hook manufacture at JRD indicates familiarity with the technological sequence, the application of different manufacturing techniques, the use of tools made of different materials, and different scales of work ranging from initial shaping to the production of very fine barbs and points. The variation in features and in technological solutions for the same need (i.e., the attachment of the line to the shaft) indicates a high level of technological sophistication, innovation and dexterity. Some of the morphological variability might reflect different individual skills within the community.

The size and shape of fish hooks and their features impacts their functionality (see above) and, consequently, the captured prey and the angling technique. In modern fishing, hooks are produced to target specific size and species of fish to maximize the fishing success. Thus, the variability in hook dimensions and functional features also suggests the intention to capture a range of sizes and species of fish with different behaviors. This reflects deep knowledge of fish behavior and suggests that the Natufian visitors to JRD intentionally targeted a broad spectrum of fish taxa with different sizes and behaviors supporting a region-wide trend toward broad exploitation of small-bodied prey at the end of the Epipalaeolithic [127–132].

## Line and hook technology

Use wear and residue data from the grooved pebbles at JRD indicates that the pebbles were intentionally modified and then affixed to a line made of soft material of botanic origin. Manufacturing marks indicate that the grooves (clearly marked except in two cases), were made using stone tools. The presence of several polish marks inside the grooves of three artifacts confirms that lines were attached to the grooves. This is further supported by the discovery of two fragments of vegetal fibers on a sample extracted from the groove of one pebble. The lines were tied to the small stones which were probably used as weights. The small size and weight of these stones (<15 g) is within the range of weights typically used for line (<30 grams; [133] rather than net fishing.

The JRD layers are littered with small limestone and basalt pebbles suitable in size and shape to function as line weights, though most were not modified by grooving. Some of the limestone pebbles show possible evidence for limited grooving, but the grooves are equivocal. The importance of the line weights to fishing at JRD is reflected in the careful selection of pebble blanks of a particular shape and size and investment in the creation of the groove around their circumference to secure the attachment weight to the line.

Although larger unmodified stones that likely served as net sinkers are abundant in both pre-Natufian and Natufian layers at JRD (especially Layer 4), with the exception of one poorly made example from Layer 5, small grooved pebbles make their first appearance in an Early Natufian context. The simultaneous appearance of the first bone hook in the earliest Natufian layer at JRD, suggest that the two acted together to provide what was already a sophisticated line and hook technology. Line weights are a crucial part of line fishing equipment as when combined with floats, they enable the bait to be positioned at the desired location underwater. Floats, such as porcupine quills [134] are typically made of perishable materials and are not typically found in archeological context. During the later stages of the Natufian, line and hook fishing continued to be frequently practiced as evident from the presence of both grooved pebbles and bone fish hooks at JRD.

## Fishing at JRD

Preliminary faunal data from JRD reveal changes in species composition and average fish body size that may be related to the appearance of line and hook technology in Natufian Layer 3c (Fig 2). In particular, fish are even more dominant in the Layer 3–0, 3a and 3c assemblages (Fig 2) than in the earlier layers at JRD or at any other Epipaleolithic site for that matter [49,55,132]. In addition, although samples are small, the proportion of larger fish in the assemblage increases in all of the Natufian assemblages (Layer 3a, 3b, 3c) in comparison to Layers 4 and 5. These changes suggest both that fishing was even more important at JRD during the Natufian than it already was in earlier layers and that hook and line technology enabled the targeted capture of larger fish than previously used fishing techniques. Nevertheless, small fish also remain abundant in all of the Natufian layers and given the location of the site next to the lakeshore and frequent inundations of the lake throughout history, it is difficult to distinguish how much of this is related to a change in the relative abundance of natural versus culturally deposited fish remains and how much can be ascribed to a change in human behavior. The difficult task of distinguishing natural from culturally deposited fish remains will require larger sample sizes, more comparative analyses, the study-of fish body-parts and ideally, the study of a natural assemblage originating close to the site (cf. [49]). A comparison of natural and cultural deposits from the site of Ohalo II located on the shore of the Sea of Galilee shows a significantly greater abundance of large Cyprinids in culturally deposited fish assemblages than those in a natural excavation 150 m from the site [49]. Thus, the greater proportion of large fish in the Natufian layers, may also suggest more significant human contribution to layer formation compared to earlier periods and thus more intensive activity at the site.

## Change in fishing technology and fishing strategies over time

Even though it is the largest collection of fish hooks for its early date, the JRD sample is small for statistical study, particularly when only complete hooks are included, or the hooks are separated by layers. Therefore, we can only suggest trends in technological evolution and fishing strategies over time. First, and most importantly, the association between fish hooks, line fishing weights and fish taxa in the Natufian layers reveals that even though there is evidence for fishing throughout the Epipaleolithic, line and hook fishing does not begin until the Natufian period. All layers at JRD are rich with evidence for fishing including fish bones, notched limestone pebbles identified as net sinkers, basalt cobbles of similar size that may have served the same purpose and a unique flint tool assemblage rich in burins and scrapers, that may also have been used for fishing and fish processing activities. Despite this, apart from the featureless hook #13, which deviates from all other hooks in its size and morphology (Figs 3 and 4; Table 2), all hooks originate from the two later Natufian layers of the site (Layers 3a and 3b) or the contact between Layer 3a and the Early Neolithic Layer 3–0. This pattern is supported by other south Levantine sites where fish hooks do not appear until the Natufian. Importantly, advanced hook technology and small weights become common at JRD only in Natufian Layer 3b. The assemblage features the first barbs and line attachment features in the region, although many of these disappear in later layers at the site. Finally, the decline in average maximum length of the fish hooks across the Natufian layers may relate to the targeting of smaller-bodied fish or a less visible shift in other fishing technologies used at the site (i.e., traps may have been used for catching larger bodied fish). Nevertheless, the size decline does not agree with the preliminary faunal analysis where fish become larger over time. Although large hooks are not very effective for capturing small fish, small hooks are capable of capturing a wide variety of fish sizes. Hook size may also relate more to the type of bait or lure chosen and the fishing conditions (smaller hooks are less likely to get tangled in vegetation and rocks) to target a particular

type of fish. These hypotheses require larger samples of fish hooks and fish bones for validation.

The appearance of line and hook technology at JRD and in the Natufian more generally, coincides with evolutionarily significant changes in human diets and associated technologies that typify the end of Epipaleolithic in this part of the world [127,128,130,132,135]. Although diets gradually broadened from the Middle Paleolithic onward in the southern Levant, this trend accelerates in the Epipaleolithic and peaks in the Natufian and following Pre-Pottery Neolithic A periods as people became more tethered to the landscape, populations grew, and local large game resources became more exhausted [127,129,136]. Not surprisingly, a parallel trend occurs in prey acquisition technology. Evidence for technology (bow and arrow, nets, traps, and hook and line technology) suitable for overcoming prey with variable characteristics also diversifies in the Epipaleolithic [4,37,52,54,57,60]. Although many of these new technologies, like the line and hook, were more complex and costly to manufacture and maintain, they ultimately lowered the cost of the acquisition of smaller-bodied, difficult to catch species like the fish from JRD. Thus, it became more efficient to forage for smaller-bodied taxa which enabled humans to better face the challenges of sedentism. It is quite expected that the diversification of prey technologies would occur hand in hand with similar changes in animal prey as humans became more proficient at exploiting new niches and prey types [137].

## Conclusion

The recovery of twenty complete and fragmentary bone fish hooks and a collection of grooved stones at JRD enabled a detailed study of their manufacture and use based on technological, use wear and residue analyses. The analyses illuminated several groundbreaking innovations in fishing technology that were not previously visible at such an early date. These include the use of inner and outer barbs, artificial baits (lures), diverse line attachment features including adhesives and stratigraphic associations between fish hooks and grooved stones. These many innovations, the variety of combinations in which they appear, and the multiple steps required to manufacture the many components (weights, hooks, lines, adhesives and lures) and then assemble them into an integrated line and hook technology attests to the emergence of complex fishing technology by the Late Epipaleolithic in which many features of modern line fishing had already appeared. This sophisticated technology attests to deep knowledge of fish behavior, ecology and acquisition strategies and fits into a larger pattern of technological and resource diversification at the end of the Pleistocene in the Levant immediately preceding the Neolithic period.

## Supporting information

**S1 Fig. Residue accumulations of smashed shell fragments: Red squares show where the residue extractions were performed.** a) #100; b) #103; c) residue extraction n. 1; d) residue extraction n. 2; e) Residue extraction n. 3; f) residue extraction n. 4. scale bars measure 500 μm.
(TIF)

**S2 Fig. Residues observed under transmitted light microscopy from extractions performed on artifacts #100 and 103.** a) Shell fragments; b) raphides; c) sediment particles and a shell fragment; d) sediment particles. Scale bars measure 20 μm.
(TIF)

**S1 Table. JRD bone fish hook measurements.**
(DOCX)

## Acknowledgments

All necessary permits were obtained for the described study, which complied with all relevant regulations. The JRD excavation is licensed by the Israel Antiquity Authority (License # G-75/2018). Thanks to O. Abel for participating in experimental fishing at JRD and C. Talsmith for researching the technology and terminology of present-day hooks. M. Kutschera provided crucial insight on the technology used in the production of the JRD bone hooks. A. Zupancich helped in the identification of the animal hair during macroscopic analysis. Finally, we thank the JRD excavation team for their meticulous work during excavation, sieving and sediment sorting which brought the JRD hooks to light.

## Author Contributions

**Conceptualization:** Antonella Pedergnana, Emanuela Cristiani, Natalie Munro, Francesco Valletta, Gonen Sharon.

**Data curation:** Antonella Pedergnana, Emanuela Cristiani, Natalie Munro, Francesco Valletta, Gonen Sharon.

**Formal analysis:** Antonella Pedergnana, Emanuela Cristiani, Natalie Munro, Francesco Valletta, Gonen Sharon.

**Funding acquisition:** Emanuela Cristiani, Natalie Munro, Gonen Sharon.

**Investigation:** Emanuela Cristiani, Natalie Munro.

**Methodology:** Antonella Pedergnana, Emanuela Cristiani, Natalie Munro, Francesco Valletta, Gonen Sharon.

**Project administration:** Antonella Pedergnana, Gonen Sharon.

**Software:** Francesco Valletta.

**Visualization:** Antonella Pedergnana, Emanuela Cristiani, Natalie Munro, Gonen Sharon.

**Writing – original draft:** Antonella Pedergnana, Emanuela Cristiani, Natalie Munro, Francesco Valletta, Gonen Sharon.

**Writing – review & editing:** Natalie Munro, Gonen Sharon.

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
