## [Decision Letter · Decision Letter 0]

6 Jul 2021

PONE-D-21-11773

Early line and hook fishing at the Epipaleolithic site of Jordan River Dureijat (Northern Israel)

PLOS ONE

Dear Dr. Sharon,

Thank you for submitting your manuscript to PLOS ONE. After careful consideration, we feel that it has merit but does not fully meet PLOS ONE’s publication criteria as it currently stands. Therefore, we invite you to submit a revised version of the manuscript that addresses the very few points raised during the review process.

We look forward to receiving your revised manuscript.

Kind regards,

Marco Peresani

Academic Editor

PLOS ONE

Journal Requirements:

3. In your manuscript, please provide additional information regarding the specimens used in your study. Ensure that you have reported specimen numbers and complete repository information, including museum name and geographic location.

For more information on PLOS ONE's requirements for paleontology and archaeology research, see https://journals.plos.org/plosone/s/submission-guidelines#loc-paleontology-and-archaeology-research.

[The JRD excavation (Israel Antiquity Authority License # G-75/2018) is supported by grants to GS from the following agencies: The Israel Science Foundation (Grant #918/17), the Brennan Foundation (two grants), the Wenner-Gren Foundation, National Geographic Society, Irene Levi Sala Care Archaeological Foundation, Tel Hai College and MIGAL research institute. This research was also supported by grants from the European Research Council(Starting Grant Project HIDDEN FOODS, grant no. 639286)awarded to EC and the National Science Foundation (BCS-1842087) awarded to NM.Analyses at TraCEr (Neuwied, Germany) were supported by the Römisch Germanisches Zentralmuseum –Leibniz Research Institute for Archeology by German Federal and Rhineland Palatinate funding(Sondertatbestand “Spurenlabor”).]

 [Research is supported by the Israel Science Foundation - www.isf.org.il (Grant #918/17) granted to G. Sharon.

This research was also supported by grants from the European Research Council - https://erc.europa.eu (Starting Grant Project HIDDEN FOODS, grant no. 639286) awarded to E. Cristiani and the National Science Foundation - https://www.nsf.gov (BCS-1842087) awarded to N. Munro.

The funders had no role in study design, data collection and analysis, decision to publish, or preparation of the manuscript.]

5. We note that Figure 1 in your submission contain map images which may be copyrighted. All PLOS content is published under the Creative Commons Attribution License (CC BY 4.0), which means that the manuscript, images, and Supporting Information files will be freely available online, and any third party is permitted to access, download, copy, distribute, and use these materials in any way, even commercially, with proper attribution. For these reasons, we cannot publish previously copyrighted maps or satellite images created using proprietary data, such as Google software (Google Maps, Street View, and Earth). For more information, see our copyright guidelines: http://journals.plos.org/plosone/s/licenses-and-copyright.

You may seek permission from the original copyright holder of Figure 1 to publish the content specifically under the CC BY 4.0 license. 

If you are unable to obtain permission from the original copyright holder to publish these figures under the CC BY 4.0 license or if the copyright holder’s requirements are incompatible with the CC BY 4.0 license, please either i) remove the figure or ii) supply a replacement figure that complies with the CC BY 4.0 license. Please check copyright information on all replacement figures and update the figure caption with source information. If applicable, please specify in the figure caption text when a figure is similar but not identical to the original image and is therefore for illustrative purposes only.

Reviewers' comments:

Reviewer's Responses to Questions

**Comments to the Author**

1. Is the manuscript technically sound, and do the data support the conclusions?

Reviewer #1: Yes

Reviewer #2: Yes

2. Has the statistical analysis been performed appropriately and rigorously? 

Reviewer #1: Yes

Reviewer #2: N/A

3. Have the authors made all data underlying the findings in their manuscript fully available?

Reviewer #1: Yes

Reviewer #2: Yes

4. Is the manuscript presented in an intelligible fashion and written in standard English?

Reviewer #1: Yes

Reviewer #2: Yes

5. Review Comments to the Author

Reviewer #1: The article presents very interesting finds of early fishing equipment. Research methods include both archaeological and scientific analyses. Results obtained during this resarch are quite persuasive. Discussion touches various sides of ancient line fishing in the Jordan River valley and in a broader context. Conclusions are clear and supported by the analyzed data. The article should be published as it is.

Reviewer #2: In summary, this is a well written, well researched, comprehensive and data rich paper that should be published with only a few minor changes. I have made some comments about useful changes.

Required change

line 125. Possible straight fish hooks (gorges or hameçon droits) were in use during the Magdalenian when artistic representations of fish appear (34), but the earliest fish hooks known thus far are made of shell and come from the cave of Jerimalai in East Timor. The site of Jerimalai has now been renamed Asitau Kura at the request of the traditional owners. Maybe it should say ' but the earliest fish hooks known thus far are made of shell and come from the cave of Asitau Kuru (formerly known as Jerimalai) in East Timor.'

Shipton, C., S. O'Connor, N. Jankowski, J. O'Connor-Veth, T. Maloney, S. Kealy and C. Boulanger 2019 A new 44,000-year sequence from Asitau Kuru (Jerimalai), Timor-Leste, indicates long-term continuity in human behaviour. Archaeological and Anthropological Sciences 11(10):5717–5741.

Line 127 Jerimalai should be changed to Asitau Kuru. Please make this change throughout and reference the Shipton et al. paper

Suggestions for useful changes or additions

Line 118. Barbed bone artefacts which are thought to have been used as projectile tips for spears are also found in East Timor in the site of Matja Kuru 2 where they are dated earlier than 30ka. As there are no large mammals on the island of Timor at this time it was argued that they may have been used in a maritime context for spearing large fish or marine animals see O’Connor, S., G. Robertson and K.P. Aplin 2014 Are osseous artefacts a window to perishable material culture? Implications of an unusually complex bone tool from the Late Pleistocene of East Timor. Journal of Human Evolution 67:108–119.

It might also be useful for the authors to look at a recent paper on the fish hook assemblage from Makpan Cave in Alor Island which is a large assemblage which also shows all stages of the production sequence as well as the items used for their production such as files.

Michelle C. Langley, Sue O’Connor, Shimona Kealy & Mahirta (2021): Fishhooks, Lures, and Sinkers: Intensive Manufacture of Marine Technology from the Terminal Pleistocene at Makpan Cave, Alor Island, Indonesia, The Journal of Island and Coastal Archaeology, DOI: 10.1080/15564894.2020.1868631

6. PLOS authors have the option to publish the peer review history of their article (what does this mean?). If published, this will include your full peer review and any attached files.

Reviewer #1: No

Reviewer #2: No

---

## [Author Response · Author response to Decision Letter 0]

5 Sep 2021

Response to Editor and Reviewer's Comments

Please see our responses to each of the reviewer and editor's comments after each suggestion below (in italics).

Editor’s Comments:

[The JRD excavation (Israel Antiquity Authority License # G-75/2018) is supported by grants to GS from the following agencies: The Israel Science Foundation (Grant #918/17), the Brennan Foundation (two grants), the Wenner-Gren Foundation, National Geographic Society, Irene Levi Sala Care Archaeological Foundation, Tel Hai College and MIGAL research institute. This research was also supported by grants from the European Research Council(Starting Grant Project HIDDEN FOODS, grant no. 639286)awarded to EC and the National Science Foundation (BCS-1842087) awarded to NM.Analyses at TraCEr (Neuwied, Germany) were supported by the Römisch Germanisches Zentralmuseum –Leibniz Research Institute for Archeology by German Federal and Rhineland Palatinate funding(Sondertatbestand “Spurenlabor”).]

 [Research is supported by the Israel Science Foundatio - www.isf.org.il (Grant #918/17) granted to G. Sharon.

This research was also supported by grants from the European Research Council - https://erc.europa.eu (Starting Grant Project HIDDEN FOODS, grant no. 639286) awarded to E. Cristiani and the National Science Foundation - https://www.nsf.gov (BCS-1842087) awarded to N. Munro.

The funders had no role in study design, data collection and analysis, decision to publish, or preparation of the manuscript.]

The funding related text was removed from the acknowledgments and the funding statement was updated. 

5. We note that Figure 1 in your submission contain map images which may be copyrighted. All PLOS content is published under the Creative Commons Attribution License (CC BY 4.0), which means that the manuscript, images, and Supporting Information files will be freely available online, and any third party is permitted to access, download, copy, distribute, and use these materials in any way, even commercially, with proper attribution. For these reasons, we cannot publish previously copyrighted maps or satellite images created using proprietary data, such as Google software (Google Maps, Street View, and Earth). For more information, see our copyright guidelines: http://journals.plos.org/plosone/s/licenses-and-copyright.

Figure 1 is comprised of a single aerial picture and 3 maps. All parts of Figure 1 were produced as part of the JRD research project and Gonen Sharon – head of the JRD excavation project, holds the rights to all parts. No use was made in any outside source (such as google earth etc.) in the creation of Figure 1.

In addition, please not that we uploaded a new corrected version of Figure 11 with corrected numbering.

Reviewer's Comments:

Reviewer #1: The article presents very interesting finds of early fishing equipment. Research methods include both archaeological and scientific analyses. Results obtained during this research are quite persuasive. Discussion touches various sides of ancient line fishing in the Jordan River valley and in a broader context. Conclusions are clear and supported by the analyzed data. The article should be published as it is.

Thank you.

Reviewer #2: In summary, this is a well written, well researched, comprehensive and data rich paper that should be published with only a few minor changes. I have made some comments about useful changes.

Required change

line 125. Possible straight fish hooks (gorges or hameçon droits) were in use during the Magdalenian when artistic representations of fish appear (34), but the earliest fish hooks known thus far are made of shell and come from the cave of Jerimalai in East Timor. The site of Jerimalai has now been renamed Asitau Kura at the request of the traditional owners. Maybe it should say ' but the earliest fish hooks known thus far are made of shell and come from the cave of Asitau Kuru (formerly known as Jerimalai) in East Timor.'

Shipton, C., S. O'Connor, N. Jankowski, J. O'Connor-Veth, T. Maloney, S. Kealy and C. Boulanger 2019 A new 44,000-year sequence from Asitau Kuru (Jerimalai), Timor-Leste, indicates long-term continuity in human behaviour. Archaeological and Anthropological Sciences 11(10):5717–5741.

The site name was changed as suggested and the reference was added

Line 127 Jerimalai should be changed to Asitau Kuru. Please make this change throughout and reference the Shipton et al. paper

These changes were made and the reference was added.

Suggestions for useful changes or additions

Line 118. Barbed bone artefacts which are thought to have been used as projectile tips for spears are also found in East Timor in the site of Matja Kuru 2 where they are dated earlier than 30ka. As there are no large mammals on the island of Timor at this time it was argued that they may have been used in a maritime context for spearing large fish or marine animals see 

O’Connor, S., G. Robertson and K.P. Aplin 2014 Are osseous artefacts a window to perishable material culture? Implications of an unusually complex bone tool from the Late Pleistocene of East Timor. Journal of Human Evolution 67:108–119.

This reference was added and the text was updated accordingly.

It might also be useful for the authors to look at a recent paper on the fish hook assemblage from Makpan Cave in Alor Island which is a large assemblage which also shows all stages of the production sequence as well as the items used for their production such as files.

Michelle C. Langley, Sue O’Connor, Shimona Kealy & Mahirta (2021): Fishhooks, Lures, and Sinkers: Intensive Manufacture of Marine Technology from the Terminal Pleistocene at Makpan Cave, Alor Island, Indonesia, The Journal of Island and Coastal Archaeology, DOI: 10.1080/15564894.2020.1868631

This paper describes another assemblage of fishhooks, part of which dates to the Late Pleistocene and also argues for the early use of lures. We added the reference and tweaked our text in several places to account for this new data. 

Thank you and all the best

Gonen Sharon

---

## [Editor Report · Decision Letter 1]

9 Sep 2021

Early line and hook fishing at the Epipaleolithic site of Jordan River Dureijat (Northern Israel)

PONE-D-21-11773R1

Dear Dr. Sharon,

We’re pleased to inform you that your manuscript has been judged scientifically suitable for publication and will be formally accepted for publication once it meets all outstanding technical requirements.

Kind regards,

Marco Peresani

Academic Editor

PLOS ONE

---

## [Editor Report · Acceptance letter]

13 Sep 2021

PONE-D-21-11773R1 

Early line and hook fishing at the Epipaleolithic site of Jordan River Dureijat (Northern Israel) 

Dear Dr. Sharon:

I'm pleased to inform you that your manuscript has been deemed suitable for publication in PLOS ONE. Congratulations! Your manuscript is now with our production department. 

Kind regards, 

on behalf of

Dr. Marco Peresani 

Academic Editor

PLOS ONE